# Agent-Oriented Planning in Multi-Agent Systems

**Ao Li**[1,2,*] **Yuexiang Xie**[3] **Songze Li**[4,5] **Fugee Tsung**[1,2] **Bolin Ding**[3] **Yaliang Li**[3,†]

[1]The Hong Kong University of Science and Technology (Guangzhou)
[2]The Hong Kong University of Science and Technology [3]Alibaba Group [4]Southeast University
[5]Engineering Research Center of Blockchain Application, Supervision and Management
(Southeast University), Ministry of Education
`aliat@connect.ust.hk, songzeli@seu.edu.cn, season@ust.hk,`
`{yuexiang.xyx,bolin.ding,yaliang.li}@alibaba-inc.com`

## ABSTRACT

Through the collaboration of multiple LLM-empowered agents possessing diverse expertise and tools, multi-agent systems achieve impressive progress in solving real-world problems. Given the user queries, the meta-agents, serving as the brain within multi-agent systems, are required to decompose the queries into multiple sub-tasks that can be allocated to suitable agents capable of solving them, so-called agent-oriented planning. In this study, we identify three critical design principles of agent-oriented planning, including solvability, completeness, and non-redundancy, to ensure that each sub-task can be effectively resolved, resulting in satisfactory responses to user queries. These principles further inspire us to propose AOP, a novel framework for agent-oriented planning in multi-agent systems, leveraging a fast task decomposition and allocation process followed by an effective and efficient evaluation via a reward model. According to the evaluation results, the meta-agent is also responsible for promptly making necessary adjustments to sub-tasks and scheduling. Besides, we integrate a feedback loop into AOP to further enhance the effectiveness and robustness of such a problem-solving process. Extensive experiments demonstrate the advancement of AOP in solving real-world problems compared to both single-agent systems and existing planning strategies for multi-agent systems. The source code is available at https://github.com/lalaliat/Agent-Oriented-Planning.

## 1 INTRODUCTION

In recent years, large language models (LLMs) (Achiam et al., 2023; Chowdhery et al., 2023; Touvron et al., 2023) have achieved impressive breakthroughs in natural language understanding and generation, marking a critical advancement in the exploration of artificial general intelligence (AGI). As the capabilities of LLMs progress, LLM-empowered agents (Qin et al., 2024; Dong et al., 2024; Wang et al., 2024) emerge as key components for integrating expertise and tools to effectively translate these advancements into practical applications. Advancing this paradigm, multi-agent systems (Shen et al., 2023; Gao et al., 2024; Wu et al., 2024), which involve multiple diverse agents, provide great flexibility and adaptability by leveraging and collaborating on the strengths of various agents, promoting comprehensive solutions to complex real-world problems.

Previous studies (Cai et al., 2023; Qian et al., 2024) propose to define suitable standard operating procedures (SOPs) based on the insights and experiences of human professionals for solving specific tasks, such as software development (Hong et al., 2024) and simulating personality traits (Serapio-García et al., 2023). In these scenarios, multiple agents are assigned to execute tasks following the predefined SOPs, resulting in remarkable successes. However, for multi-agent systems designed to tackle a diverse range of complex real-world problems, there is a critical need for a central entity that can automatically generate task-specific operating procedures and coordinate the activities of various agents (Wang et al., 2024).

This central entity, typically referred to as a meta-agent (a.k.a. a controller or planner), carries two primary responsibilities. Firstly, the meta-agent needs to comprehend user queries and decompose

---

*Work done as an intern at Alibaba Group.

†Corresponding author.

them into several sub-tasks, ensuring that each sub-task can be adequately addressed by a single agent. Secondly, the meta-agent is also expected to assign these sub-tasks to the appropriate agents for execution, enabling the solutions of these sub-tasks to collectively provide comprehensive answers to the original user queries.

However, since the meta-agent cannot effectively associate sub-tasks with agents based on the provided agents descriptions (Liu et al., 2024; Hong et al., 2024), the performance of task decomposition and allocations might be sub-optional. To tackle these challenges, we identify three design principles for an agent-oriented planning framework, including solvability, completeness, and non-redundancy. These principles further inspire us to propose **AOP**, a novel framework for **A**gent-**O**riented **P**lanning in multi-agent systems.

Specifically, to resolve a user query, the meta-agent first performs fast task decomposition and allocation, serving as intermediate results that can be further modified and revised. The proposed AOP incorporates a reward model designed to efficiently evaluate the solvability of sub-tasks without requiring actual agent calls. According to the evaluation results, some sub-tasks may be executed by the assigned agents, others may be deemed inappropriate and require re-planning, while the remaining sub-tasks are further assessed using the representative works mechanism, which helps determine whether they should be planned in detail or re-described for better aligning the ability of agents. Meanwhile, a detector is utilized to identify the missing key information or redundant content in the decomposed sub-tasks, and to provide suggestions to the meta-agent for enhancing the completeness and non-redundancy. Besides, a feedback loop is integrated into AOP to promote ongoing enhancements for the problem-solving process.

Extensive experiments are conducted based on several reasoning datasets that require collaboration among multiple LLM-empowered agents. Comparisons between AOP and baseline methods demonstrate the remarkable advancements achieved by the proposed framework. Furthermore, we conduct an ablation study to show the contributions of different components in AOP, and provide discussions on the potential for further enhancing agent-oriented planning within multi-agent systems.

## 2 RELATED WORK

Large language models (LLMs) (Brown et al., 2020; Achiam et al., 2023; Chowdhery et al., 2023; Touvron et al., 2023; Le Scao et al., 2023; Zhang et al., 2022) have demonstrated remarkable capabilities in understanding and generating human language (Zhao et al., 2024). To enhance LLMs' ability to solve complex problems, some research focuses on improving the internal reasoning capabilities of LLMs, typically involving decomposing complex questions into sequential intermediate steps before generating the final responses, as exemplified by Chain-of-Thought (CoT) prompting (Wei et al., 2022) and its variants (Kojima et al., 2022; Wang et al., 2023; Zhou et al., 2023; Li et al., 2023). For further improvements, recent studies propose to adopt multi-agent systems (Wu et al., 2024; Hong et al., 2024; Wu et al., 2025) which leverage the collective intelligence and specialized skills of multiple LLM-empowered agents to solve tasks collaboratively.

Traditional cooperative multi-agent planning methods mainly rely on symbolic methods or reinforcement learning-based methods (He et al., 2016; Yao et al., 2020), with MA-PDDL (Kovacs et al., 2012) being a key example, establishing a standard specification language for multi-agent planning. However, these methods can be limited by their complexity and heavy reliance on human experts (Huang et al., 2024). Recently, the development of LLMs helps to simplify and optimize the planning process through the creation of meta-agents, which serve as controllers that manage the flow of operations, to perform task decomposition (Huang et al., 2024). There are mainly two technical approaches: interleaved decomposition and decomposition-first. Regarding the first kind of approach, React (Yao et al., 2023) generates reasoning and action in an interleaved manner, where reasoning helps the meta-agent update action plans and action enables interaction with agents. HUSKY (Kim et al., 2024) trains an action model to iterate between two stages: generating the next action to take and executing the action using expert models and updating the current solution state. Given that the excessively long trajectories might bring hallucinations in interleaved decomposition, HuggingGPT (Shen et al., 2023) addresses this issue by carefully designing prompts to instruct ChatGPT to decompose tasks, selecting models in HuggingFace and summarizing the generated responses. Chameleon (Lu et al., 2023) prompts the meta-agent with tool descriptions and usage

examples to infer a program composed of a sequence of tools to execute for generating the final response.

Although remarkable progress has been made, plans generated solely through prompt-based methods often fall short of meeting the principles of solvability, completeness, and non-redundancy issues that have largely been overlooked in previous studies, which motivates us to propose a novel agent-oriented planning framework for multi-agent systems.

## 3 PRELIMINARIES

**Definition of Agent-Oriented Planning** In this study, a multi-agent system involves a series of LLM-empowered agents for solving real-world problems. These agents are constructed by providing specialized tools and unique system prompts that set their identities and instructions, and are powered by LLMs for query understanding, tool use, and response generation. For example, a search agent can utilize search engines to retrieve up-to-date information relevant to a given query, while a code agent is capable of generating code and executing via a code interpreter. The collaboration among these agents is facilitated by a meta-agent, which serves as the brain of the multi-agent system. When a user query is submitted, the meta-agent sends the relevant requests to the appropriate agents, optimizing for both effectiveness and efficiency. The responses produced by these agents are finally aggregated and synthesized to generate a comprehensive answer to the original user query.

Formally, given a user query $Q$, a meta-agent $\mathcal{P}$ needs to select a set of appropriate agents from a total of $n$ agents, denoted as $\mathcal{A} = \{\mathcal{A}_1, ..., \mathcal{A}_n\}$. Each agent is associated with a description $d$ that outlines its capability. We denote the collection of all agent descriptions as $\mathcal{D} = \{d_1, ..., d_n\}$. The user query $Q$ can be decomposed by $\mathcal{P}$ into $m$ sub-tasks, with each sub-task assigned to a specific agent based on their descriptions:

$$\mathcal{P}(Q, \mathcal{D}, \mathcal{A}) = \{(q_i, \mathcal{A}'_i) \mid i \in [m]\}. \tag{1}$$

After that, each selected agent $i \in [m]$ produces a response to its assigned sub-task, denoted as $r_i = \mathcal{A}'_i(q_i)$. These responses $\{r_1, ..., r_m\}$ are then utilized to generate the final answer to resolve the original query $Q$. Such a process is called *agent-oriented planning*.

**Challenges** The challenges of agent-oriented planning in multi-agent systems can be two-fold. Firstly, different from existing studies focused on task decomposition (Shen et al., 2023; Lu et al., 2023) or chain-of-thought reasoning (Wei et al., 2022), agent-oriented planning requires intentional decomposition of user queries to effectively associate sub-tasks with agents, which includes considerations of the description of sub-tasks, the granularity of decomposition, the format of the responses, and so on. An example is illustrated at the higher left of Figure 1. Given a user query "How much tin (kg) with 100 kg copper can lower the mixture's melting point to 800 ℃?", a naive decomposition might suggest "Determine the melting point of tin and copper" followed by "Calculate the amount of tin (kg) required to reduce the melting point of the mixture to 800 ℃ with 100 kg copper". However, when the sub-task of determining the melting points is assigned to a search agent, it may not result in satisfied responses since the query involves two entities simultaneously. In the context of agent-oriented planning, it is important to consider the capabilities of agents. As a result, such a sub-task should be further decomposed into individual entity searches, i.e., first determine the melting point of tin and then determine the melting point of copper.

Secondly, assigning sub-tasks to appropriate agents is non-trivial, as the meta-agent can only rely on the agents' descriptions to determine task allocation in most cases (Shen et al., 2023). However, a concise and highly generalized description that adequately illustrates an agent's capabilities may not always be available, leading to suboptimal allocation. For example, if the description of a commonsense agent does not specify the extent of its knowledge base, the meta-agent may struggle to ascertain whether it is suitable to assign the sub-task of querying the melting points of tin and copper to that agent, as shown at the lower left of Figure 1.

**Design Principles** We further identify three critical principles that guide the design of AOP for effective and efficient agent-oriented planning in multi-agent systems:

- *Solvability*. Each sub-task $q_i$, $\forall i \in [m]$ should be independently and completely resolvable by at least one single agent within the multi-agent system, ensuring that the response for

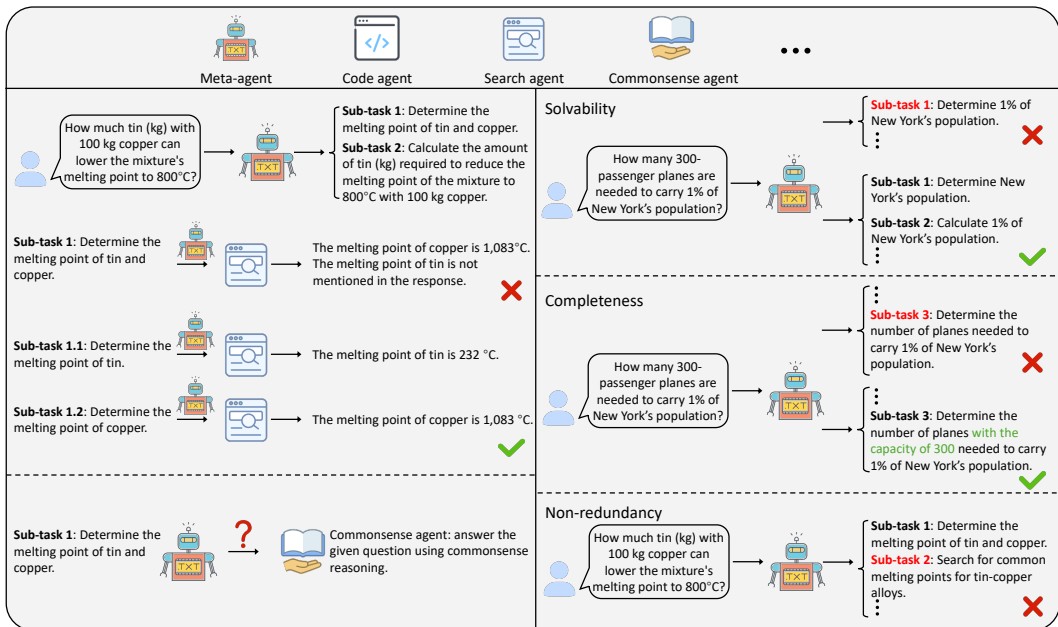

Figure 1: Examples of agent-oriented planning in multi-agent systems, regarding two challenges (left side) and three design principles (right side).

each sub-task can be reliable. If a sub-task does not satisfy solvability, the meta-agent is expected to take some modifications or further decomposition.

- *Completeness*. The array of sub-tasks $\{q_1, ..., q_m\}$ should include all necessary information from the original user query $Q$, which ensures that the aggregation of responses of these sub-tasks can effectively yield a comprehensive answer to the user query. While a sub-task might include only part of the necessary information, it is not allowable for any particular piece of critical information to be omitted from all sub-tasks.

- *Non-Redundancy*. The array of sub-tasks $\{q_1, ..., q_m\}$ should not include redundant elements, avoiding those task executions that are either irrelevant to resolving $Q$ or duplicated. The principle of non-redundancy promotes that the sub-tasks form a minimal effective set necessary to address the user query, enhancing overall efficiency.

Several examples are provided on the right side in Figure 1 for a better understanding of these design principles. Note that here we focus on the redundancy between sub-tasks that should be simplified, and believe that adding redundancy among agents is necessary for fault tolerance as discussed in Appendix B, especially considering unexpected changes in agent availability.

While these principles may appear general in nature, we propose an explicit specification and systematic design that follows these principles in the domain of LLM-based multi-agent systems. We hope that these principles, along with the framework we have outlined in the next section, can serve as a good starting point and inspire further research in this field.

## 4 AGENT-ORIENTED PLANNING IN MULTI-AGENT SYSTEMS

In this section, we introduce the details of the proposed AOP framework, with the overall architecture illustrated in Figure 2.

### 4.1 FAST DECOMPOSITION AND ALLOCATION

First of all, following existing studies (Shen et al., 2023; Chen et al., 2024), we provide detailed instructions to the meta-agent to guide it in performing agent-oriented planning. What sets our approach apart is that the provided instructions incorporate the following requirements: (i) Integration

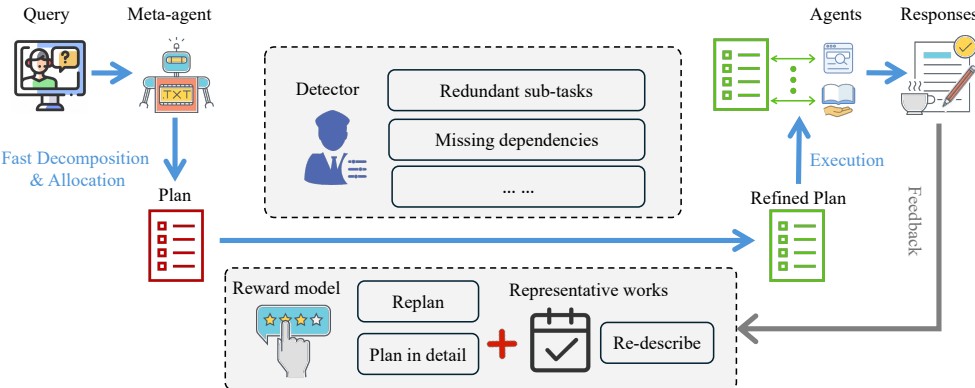

Figure 2: Overall architecture of AOP. The framework begins with the *meta-agent* performing a fast decomposition and allocation of the received user query, resulting in an initial plan. A *detector* is employed to improve the completeness and eliminate redundancy of the plan, while a *reward model* provides scores to guide the meta-agent in refining the plan further, which involves operations such as replan, plan-in-detail, etc. The refined plan is sent back to the multi-agent system for generating responses to the user query.

of the user query $Q$ and all the agent descriptions $\mathcal{D}$: We include both the user query and the agents' descriptions in the instructions, promoting the meta-agent to fully consider the capabilities of each agent and tailor the sub-tasks to align with these capabilities. (ii) Suggestions for assigned agents: We require the meta-agent to provide suggestions for the agents assigned to each decomposed sub-task. Although we tend to separate task decomposition and assignment into two independent tasks performed sequentially, the experimental observations indicate that combining these two tasks enhances the effectiveness of agent-oriented planning. (iii) Structured decomposition of tasks: The decomposed tasks are required to be structured in a sequential manner, specifying any dependencies that may exist between sub-tasks, which ensures that the execution of tasks follows a correct logical order. The adopted prompt can be found in Appendix A.1.

The aforementioned process of agent-oriented planning, which is called *fast decomposition and allocation* in this study, heavily relies on the capabilities of the meta-agent and meticulously designed system prompts. While such an end-to-end approach can achieve high efficiency, its success rate, namely fulfilling the three design principles mentioned above and providing reliable answers to the original user query, and its stability, namely the meta-agent can follow instructions and produce formatted responses, might not be satisfactory.

Inspired by recent studies on inference time computing (OpenAI, 2024), we propose to design mechanisms to guide the meta-agent toward more comprehensive reasoning processes. The results of fast decomposition and allocation should be viewed as intermediate outputs rather than final results, which need to be further evaluated to offer detailed revision suggestions for the meta-agent. More details on this proposed approach are provided in the next subsection.

## 4.2 IS A SUB-TASK COMPLETELY RESOLVED?

To determine whether a sub-task can be resolved by a single agent, i.e., satisfying the principle of solvability, a straightforward solution is to send the sub-task to every agent in the system. Each agent generates a response to the sub-task, allowing us to select the best answer and assess whether it completely resolves the sub-task. However, this approach is often impractical due to the unafford-able overhead. For a user query decomposed into $m$ sub-tasks, a multi-agent system with $n$ agents would need to execute a total of $m \times n$ agent calls for just a single trial, making it an inefficient strategy in scenarios with a large number of agents and sub-tasks.

To tackle this, we introduce a reward model that provides efficient evaluations of the solvability of sub-tasks, which aims to predict the quality of the agents' responses to sub-tasks without necessitating actual agent calls.

Specifically, we first prepare a dataset for training the reward model. For a given user query $Q$, we follow the fast decomposition and allocation process introduced in Section 4.1, requiring the meta-agent to decompose $Q$ into several sub-tasks and select $l$ agents for each sub-task based on the agents' descriptions, which can be formally given as follows:

$$\mathcal{P}(Q, \mathcal{D}, \mathcal{A}) = \{(q_i, \mathcal{A}_{i,1}, ..., \mathcal{A}_{i,l}) \mid i \in [m]\}. \tag{2}$$

After that, we execute the plan, i.e., sending the sub-tasks to the assigned agents, and obtain all the responses from agents as:

$$\{r_{i,j} = \mathcal{A}_{i,j}(q_i) \mid i \in [m], j \in [l]\}, \tag{3}$$

where the choice of $l$ can be a trade-off. A large $l$ leads to comprehensive responses from lots of agents for constructing the training dataset, while it might need more computation resources and affect the overall quality of the dataset. In this study, we set $l$ to be half the number of agents in the multi-agent systems.

We utilize a scorer $\mathcal{S}$ that evaluates agents' responses to sub-tasks, i.e., $\mathcal{S}(q_i, r_{i,j}) = s_{i,j}$, which serves as annotations in the training dataset. The scorer $\mathcal{S}$ provides evaluations from three key aspects, including correctness, relevance, and completeness, and can be implemented by using LLMs or employing human annotators. We apply this scoring process across a diverse array of user queries $\mathcal{Q} = \{Q_k\}_{k=1}^K$ and collect the results to form the training dataset, denoted as $\mathcal{T} = \{(q_{k,i}, d_{k,i,j}, s_{k,i,j}) \mid k \in [K], i \in [m_k], j \in [l]\}$. More details on the construction of the dataset, such as the adopted prompts, are summarized in Appendix C.

The reward model $\mathcal{M}$, parameterized as $\theta$, consists of embedding layers followed by fully connected layers. We obtain embeddings for both the sub-task and the agent's description separately, then concatenate these embeddings together and feed them into the fully connected layers for further processing. To provide evaluations without making actual agent calls, we design the training objective aimed at minimizing the discrepancies between the model's predictions and the annotations provided by the scorer, which can be formulated as follows:

$$L(\mathcal{T}, \theta) = \frac{1}{K} \sum_{k=1}^K \frac{1}{m_k} \sum_{i=1}^{m_k} \frac{1}{l} \sum_{j=1}^{l} (s_{k,i,j} - \mathcal{M}_\theta(q_{k,i}, d_{k,i,j}))^2. \tag{4}$$

With the well-trained reward model, the evaluation of the results produced by fast decomposition and allocation can be summarized as follows. For each suggestion provided by the meta-agent, stating the assignment of sub-task $q_i$ to agent $\mathcal{A}_i'$, the reward model predicts a score denoted as $\mathcal{M}_\theta(q_i, d_i')$. A sufficiently high predicted score, which implies that $q_i$ can be completely resolved by agent $\mathcal{A}_i'$, leads to the acceptance of this suggestion. Conversely, if the predicted score falls below a predefined threshold, which indicates that the assignment should be revisited by the meta-agent, the reward model provides a set of scores $\hat{s}_{i,j} = \mathcal{M}_\theta(q_i, d_j)$ for all the agents $\mathcal{A}_j, j = 1, ..., n$. From these scores, the meta-agent reports the optimal selection as $j_{max} = \arg\max_j \hat{s}_{i,j}$.

Note that there may be scenarios where the highest score $\hat{s}_{i,j_{max}}$ is still extremely low, indicating that the sub-task $q_i$ does not satisfy the principle of solvability and that no single agent can resolve it independently. In these cases, the meta-agent $\mathcal{P}$ is required to perform a *replan* on $q_i$. The prompt used for replan is detailed in Appendix A.2.

## 4.3 SUB-TASK MODIFICATIONS

In addition to the two scenarios mentioned in Section 4.2, where the assignment of sub-task $q_i$ to agent $\mathcal{A}_i'$ is either predicted to be acceptable by the reward model or to be entirely inappropriate and requires a replan process, there also exists another situation where agent $\mathcal{A}_i'$ might not provide a reliable response to sub-task $q_i$, even though the assignment seems reasonable according to the reward model.

We identify two intrinsic reasons for such a situation in agent-oriented planning. First, the description of the sub-task may lack critical information necessary for solving the problem, such as the interpretation of contextual elements like pronouns or other ambiguous terms. In such cases, while the agent possesses the capability to resolve the sub-task, the missing information can hinder it from providing a satisfactory response. Second, the sub-task $q_i$ might be too complex, therefore a single

agent can only address parts of this sub-task based on its expertise and tools. In this subsection, we design mechanisms to distinguish between these two cases and propose corresponding solutions.

To be more specific, for each agent, we propose to construct a set of representative works that record the sub-tasks it has completely resolved. We would initially bypass the construction and update for these representative works (refer to Section 4.5 for more details) and focus on how to utilize them to make sub-tasks modifications here. Given a sub-task $q_i$, we calculate its similarity with a representative work $q_t$ following $cos(\mathcal{E}(q_i), \mathcal{E}(q_t))$, where $\mathcal{E}$ is the embedding part in $\mathcal{M}$ and $cos(u, v) = \frac{u \cdot v}{\|u\|_2 \|v\|_2}$ calculates the cosine similarity. We define the similarity as follows:

$$sim(q_i, \mathcal{Q}_j) = \max\{cos(\mathcal{E}(q_i), \mathcal{E}(q_t))|q_t \in \mathcal{Q}_j\}, \quad (5)$$

where it can be denoted as $sim_{i,j}$ for short.

A large $\max\{sim_{i,j} \mid j \in [n]\}$ indicates that the representative works of the agent corresponding to $\max\{sim_{i,j} \mid j \in [n]\}$ contain a sub-task similar to $q_i$. Such a case should be attributed to the first reason mentioned above, and we tend to request the meta-agent to perform *re-describe* on $q_i$ according to the similar representative work.

On the other hand, when the calculated similarity does not meet the threshold, we regard $q_i$ as too complex for any single agent in the system to solve (corresponding to the second reason above). As a result, the meta-agent $\mathcal{P}$ is requested to perform *plan-in-detail* for further decomposing the sub-task into simple ones. To avoid creating redundant sub-tasks, we provide $\mathcal{P}$ with all the sub-tasks and explicit instruction on avoiding overlapping sub-asks. The prompt used for sub-task modification can be found in Appendix A.3 and Appendix A.4.

## 4.4 DETECTOR FOR COMPLETENESS AND NON-REDUNDANCY

Following the idea in fast decomposition and allocation, we attempt to incorporate detailed instructions to the meta-agent to satisfy the principles of completeness and non-redundancy. However, empirical observations reveal that these instructions may not always work well. In fact, more than 15% of user queries still exhibit such issues during decomposition in our initial experiments.

To tackle these issues, we involve a detector, implemented by providing a role-play prompt to LLMs, to evaluate both the completeness and non-redundancy of the intermediate results provided by the meta-agent. For the principle of completeness, the detector extracts all key elements and requirements from the original query and then matches these elements against each sub-task, determining whether the sub-tasks collectively address all essential aspects of the original task. To further ensure that each sub-task can be successfully executed, the detector is also required to determine whether there are any additional dependencies needed beyond those identified during decomposition (e.g., unforeseen dependencies). If such dependencies exist, the execution results of these dependent sub-tasks should also be provided as inputs.

Regarding non-redundancy, the detector is responsible for analyzing whether any two subtasks contain identical information or try to solve the same problem, and whether there are unnecessary subtasks that do not contribute to the user query's solution.

When the provided results do not satisfy the principles of completeness or non-redundancy, the detector identifies the missing key information or redundant content and offers recommendations for refining the plan, such as suggestions for supplementing missing details or removing overlapping sub-tasks. The prompt used for the detector is provided in Appendix A.5. From the experimental results shown in Table 2, we observe that the detector effectively mitigates the issues of incompleteness and redundancy, leading to an effective and efficient agent-oriented planning process.

## 4.5 FEEDBACK LOOP

An automatic feedback loop is integrated into AOP, promoting the ongoing enhancement of the problem-solving process. As introduced in Section 4.2, the reward model can identify certain sub-tasks as being resolvable by specific agents. These identified sub-tasks are then collected to form the *representative works* of the corresponding agents. The training data for the reward model, which includes the sub-tasks and the ground truth scores of responses provided by the agents, is utilized to initialize these representative works.

Note that the representative works are continuously updated. When a user query is completely resolved, the sub-tasks decomposed from that user query can be added to the representative works of the agents who are selected to provide responses to these sub-tasks. To prevent redundancy in the representative works, we implement a similarity threshold. This threshold ensures that only new and sufficiently distinct sub-tasks are incorporated into an agent's representative works, maintaining the diversity and relevance of the tasks that each agent has previously resolved.

## 5 EXPERIMENTS

In this section, we provide empirical comparisons between AOP and the existing studies.

### 5.1 SETTINGS

**Datasets & Evaluations** We conduct experiments based on a numerical reasoning dataset (Kim et al., 2024), which necessitates the collaboration of multiple agents in resolving the queries. For example, a query can be "If Sarah wants to buy one BMW X5 and one Tesla Model 3, how much more would she need to pay to buy the BMW X5 compared to the Tesla Model 3?" To resolve this query, we first need to search for the prices of the BMW X5 and Tesla Model 3, and then calculate the price difference between them. Following previous study (Kim et al., 2024), we adopt Husky-QA, which consists of 1,440 queries in the training data and 292 queries in the test data. Besides, we also provide more experimental results on the decontextualized versions of a subset of DROP (Dua et al., 2019) and IIRC (Ferguson et al., 2020) in Appendix D.1.

For quantification comparisons, we provide instructions to GPT-4o (OpenAI, 2024) for assisting in judging whether the execution results align with the ground truth and calculate accuracy as the evaluation metric. The prompt used for evaluation can be found in Appendix A.6.

**Involving Agents** In the experiments, the multi-agent system includes a meta-agent and several diverse agents, including a code agent, math agent, search agent, and commonsense agent. The descriptions of these agents, which are similar to those in previous studies (Kim et al., 2024) for a fair comparison, are shown below:

- *Code Agent*: Generate code in Python for precise computations to solve the given task.
- *Math Agent*: Answer math questions by reasoning step-by-step.
- *Search Agent*: Call the Bing Search API to obtain information related to the given task.
- *Commonsense Agent*: Answer the given question using commonsense reasoning.

The meta-agent is tasked with decomposing user queries into sub-tasks and assigning the most suitable agents to execute those sub-tasks. We use GPT-4o as the employed LLM for all agents, and the prompts used are provided in Appendix A.7. Besides, an investigation on the effectiveness of involving expert agents powered by task-specified LLMs, and the method for efficiently handling the coexistence of similar agents can be found in Appendix E.

**Reward Model** We adopt all-MiniLM-L6-v2 (Wang et al., 2020) as the embedding layers of the reward model, which maps text sentences or paragraphs into a 384-dimensional dense vector space. The embeddings for both the sub-task and the agent's description are computed separately, and then are concatenated together to form a 768-dimensional dense vector. Following the embedding model, we add a three-layer multilayer perceptron (MLP) with output dimensions of 256,64, and 1, respectively. We freeze the embedding layers and only fine-tune the parameters of the MLP, making the training process efficient. The batch size is set to 32, and the learning rate is 1e-3. We train the reward model for 50 epochs on one Tesla V100-SXM2-32GB GPU.

**Baselines** For the single-agent systems, we utilize **GPT-4o** as a baseline method, providing user queries directly without any additional prompt engineering. Besides, we provide GPT-4o with Chain-of-Thought (CoT) (Wei et al., 2022) prompt and Zero-Shot CoT (Kojima et al., 2022) prompt to guide the LLM to perform reasoning processes, resulting in two baselines denoted as **CoT** and **Zero-Shot CoT**.

Table 1: Comparisons between AOP and baselines.

| Method | Accuracy (%) | Prompt Tokens (M) | Completion Tokens (M) | Time (s) |
|---|---|---|---|---|
| GPT-4o | 33.3 | 0.02 | 0.09 | 1,968 |
| CoT | 35.6 | 0.07 | 0.08 | 1,375 |
| Zero-Shot CoT | 32.2 | 0.02 | 0.11 | 1,565 |
| Meta-Agent | 30.0 | 0.65 | 0.13 | 5,364 |
| Meta-Agent: Traversal | 35.2 | 3.07 | 0.69 | 23,175 |
| REACT | 37.6 | 2.47 | 0.19 | 11,510 |
| HUSKY | 39.6 | 0.83 | 0.15 | 10,394 |
| AOP (ours) | **43.7** | 1.12 | 0.38 | 11,869 |

Regarding the multi-agent systems, we instruct GPT-4o to serve as a meta-agent responsible for performing fast task decomposition and allocation. We implement two baselines, denoted as **Meta-Agent** and **Meta-Agent: Traversal**, respectively. Meta-Agent denotes that GPT-4o selects one agent for each sub-task, while Meta-Agent: Traversal denotes that GPT-4o iteratively queries all agents for each sub-task and selects the best response. Besides, we compare AOP with **REACT** (Yao et al., 2023) and **HUSKY** (Kim et al., 2024), which are representative methods involving task decomposition and sub-task execution.

## 5.2 COMPARISONS AND ANALYSIS

The comparisons between AOP and the baseline methods are shown in Table 1. We adopt accuracy as the metric for comparing effectiveness, and use prompt/completion tokens (refer to the total cost on the whole test dataset) and the execution time for comparing efficiency.

Overall, the experimental results demonstrate that AOP achieves notable improvements compared to all baseline methods. To be specific, when comparing with single-agent systems including GPT-4o, CoT, and Zero-Shot CoT, AOP outperforms these systems by 10.4%, 8.1%, and 11.5% in terms of accuracy, respectively. These improvements can be attributed to the collaboration among multiple diverse agents and the effective scheduling provided by the meta-agent.

It is not surprising to observe that the cost of AOP is significantly higher than that of single-agent systems, and is at the same level compared to other multi-agent systems. These additional costs during the inference phase, which include task decomposition, allocation, and modifications carried out by the meta-agent, are affordable and can be worthwhile as long as they bring significant improvements in accuracy and stability when applied to real-world applications. Such an exploration aligns with a recent study (OpenAI, 2024) in inference time computing, aimed at efficiently utilizing more tokens to resolve challenging tasks.

Compared to systems that also involve a meta-agent for task decomposition and allocation, we can observe from the table that AOP achieves at least a 4.1% improvement in terms of accuracy while maintaining the same level of computation costs and inference times. The results of Meta-Agent and Meta-Agent: Iteration indicate that simply instructing GPT-4o to perform task decomposition and allocation does not always yield satisfactory responses, even though the capabilities of LLMs are recognized as powerful. Existing studies in task decomposition and allocation fail to consider the abilities and characteristics of the agents beyond their descriptions, and lack mechanisms for modifications and feedback within the multi-agent system, which can be hindered by the challenges as summarized in Section 3. AOP is built upon these identified design principles and incorporates novel mechanisms for automatically evaluating intermediate results, allowing timely modifications and revisions, which guide and leverage the strengths of agents in the system to effectively resolve user queries and enhance the overall system performance.

## 5.3 FURTHER DISCUSSIONS

**Ablation Study** We conducted an ablation study to confirm the contributions of different components in AOP. Specifically, we disable the detector, the reward model, and the representative works of agents in separate experiments, with results reported in Table 2. From these results, we can

Table 2: Experimental results of ablation study.

| Method | Accuracy (%) | Prompt Tokens (M) | Completion Tokens (M) | Time (s) |
|---|---|---|---|---|
| AOP | 43.7 | 1.12 | 0.38 | 11,869 |
| w/o Plan Detector | 36.6 | 1.05 | 0.29 | 9,504 |
| w/o Reward Model | 38.7 | 1.17 | 0.37 | 11,651 |
| w/o Representative Works | 41.1 | 1.14 | 0.36 | 11,178 |

Table 3: Experimental results on the impact of scorer and reward model

| Method | Accuracy (%) | Prompt Tokens (M) | Completion Tokens (M) | Time (s) |
|---|---|---|---|---|
| AOP | 43.7 | 1.12 | 0.38 | 11,869 |
| + Manual Scoring | 46.9 | 1.26 | 0.40 | 12,676 |
| + Full Parameter Tuning | 47.9 | 1.28 | 0.41 | 13,063 |

observe that the detector has a significant impact on execution accuracy, as indicated by a notable increase in the incompleteness rate during the meta-agent's task decomposition phase. Both the reward model and the representative works are necessary for ensuring the solvability of sub-tasks and for selecting the most suitable agents for each sub-task. Overall, these results demonstrate that all three components are indispensable, working together to ensure the feasibility of task decomposition and allocation, leading to satisfactory responses to the original user queries.

**Impact of the Scorer and Reward Model**   The training dataset is constructed based on annotations provided by the scorer, which in previous experiments was implemented using LLMs. In this section, we shift to using human annotators to manually provide scores for the responses, evaluating whether a reward model trained on this manually labeled dataset would further enhance performance. This approach is denoted as **Manual Scoring**. More detailed information about the manual scoring process can be found in Appendix C. Besides, based on the manually scored training dataset, we investigate the setting of updating the embedding layers of the reward model, denoted as **Full Parameter Tuning**.

The experimental results shown in Table 3 indicate that both manual scoring and full parameter tuning lead to improved performance. These results suggest the potential for further improving AOP by providing high-quality datasets for robust training. Although using a human-expert-based scorer can lead to further improvements, we predominantly adopt a model-based scorer in the above experiments, as it is more cost-effective and generalizable. Besides, we provide empirical studies on the generalization capability of the reward model in Appendix D.2, showing that the reward model trained on one dataset demonstrates good generalization to other datasets.

## 6   CONCLUSION

In this study, we propose AOP, a novel agent-oriented planning framework for multi-agent systems, following three critical design principles to ensure that the meta-agent can effectively decompose the user query into several sub-tasks for producing satisfactory responses. AOP utilizes a fast decomposition and allocation process, which relies on the ability of LLMs to generate an intermediate schedule efficiently. After that, a reward model and the representative works mechanism are employed to evaluate these intermediate results, resulting in executing the sub-task or making necessary modifications to align the sub-task with agents, such as replan the sub-task, plan in detail, and redescribe. Extensive experiments demonstrate that AOP achieves significant improvements over both the existing single-agent and multi-agent baseline methods. We provide discussions on the contributions of different components in AOP and the potential for improvements in agent-oriented planning. We hope AOP can contribute to advancing research in this field, inspiring further exploration and innovation in practical applications.

ACKNOWLEDGMENTS

This work is funded by National Natural Science Foundation of China Grant No. 72371217, the Guangzhou Industrial Informatic and Intelligence Key Laboratory No. 2024A03J0628, the Nansha Key Area Science and Technology Project No. 2023ZD003, and Project No. 2021JC02X191, and the Fundamental Research Funds for the Central Universities Grant No. 2242024k30059.

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

# A PROMPTS USED IN AOP

## A.1 META-AGENT

The one-shot prompt for meta-agent's fast decomposition and allocation is shown in Figure 3.

```
meta_agent_prompt = '''
You are a planning agent. You are responsible for decomposing the given query into sub-tasks
    and choosing the most suitable agent for each sub-task. Your main goal is to efficiently
    and accurately complete task planning based on the descriptions of agents provided,
    ensuring the coherence and quality of the sub-tasks.
Please output the sub-tasks and corresponding agents in the following JSON format: [{"task":
    task_description, "id": task_id, "name": name_of_agent, "reason":
    your_detailed_reason_for_the_choice, "dep":dependency_task_ids}]. In this format, "task"
    is the description of the sub-task, which will be used as the input of the chosen agent;
    "dep" denotes the id of the previous sub-task which generates a new resource relied by
    the current sub-task.
The available agents and the corresponding descriptions are: [code_agent: Generate code in
    Python for precise computations to solve the given task. math_agent: Answer math
    questions by reasoning step-by-step. search_agent: Call Bing Search API to obtain
    information regarding the given task. commonsense_agent: Answer the given question using
    commonsense reasoning.].
---
Here is an example:
User query: If a plane can carry 300 passengers and decides to fly from China to Indonesia,
    how many full flights are needed to transport 1\% of the population of China to Indonesia
    ?
Output:
[
    {
        "task": "Determine the population of China.",
        "id": 1,
        "name": "search_agent",
        "reason": "The search_agent can find the most recent and accurate population data for
            China.",
        "dep": []
    },
    {
        "task": "Calculate 1% of the population of China.",
        "id": 2,
        "name": "math_agent",
        "reason": "The math_agent can reason through the calculation step-by-step.",
        "dep": [1]
    },
    {
        "task": "Determine the number of full flights needed to transport 1% of the population
            of China to Indonesia, given that each plane can carry 300 passengers.",
        "id": 3,
        "name": "math_agent",
        "reason": "The math_agent can reason through the division and rounding process.",
        "dep": [2]
    }
]
---
Given the user query, output the task plan with the format above directly. Make sure all the
    important information such as nouns or numbers are included in the sub-tasks. If you
    think the query can be done with just one agent, you can output only one sub-task.
'''
```

Figure 3: One-shot prompt for the meta-agent.

## A.2 REPLAN

The replan prompt is shown in Figure 4.

```
replan_prompt = '''
You are a planning agent. You are preliminarily responsible for decomposing the given query
    into sub-tasks and choose the most suitable agent for each sub-task according to the
    following json format: [{"task": task_description, "id": task_id, "name": name_of_agent,
    "reason": your_detailed_reason_for_the_choice, "dep":dependency_task_ids}]. In this
    format, "task" is the description of the sub-task, which will be used as the input of the
     chosen agent; "dep" denotes the id of the previous sub-task which generates a new
    resource relied by the current sub-task.
The available agents and the corresponding descriptions are: [code_agent: Generate code in
    Python for precise computations to solve the given task. math_agent: Answer math
    questions by reasoning step-by-step. search_agent: Call Bing Search API for obtaining
    information regarding the given task. commonsense_agent: Answer the given question using
    commonsense reasoning.].
Given the user query: %s, the preliminary task decomposition is: %s.
But the sub-task: %s cannot be solved by any agent. Now you are responsible for replaning this
     sub-task based on agents' capabilities. Output the new sub-task with the format above
    directly.
'''
```

Figure 4: Replan prompt.

## A.3 PLAN IN DETAIL

The prompt for planning in detail is shown in Figure 5.

```
plan_in_detail_prompt = '''
You are a planning agent. You are preliminarily responsible for decomposing the given query
    into sub-tasks and choose the most suitable agent for each sub-task according to the
    following json format: [{"task": task_description, "id": task_id, "name": name_of_agent,
    "reason": your_detailed_reason_for_the_choice, "dep":dependency_task_ids}]. In this
    format, "task" is the description of the sub-task, which will be used as the input of the
     chosen agent; "dep" denotes the id of the previous sub-task which generates a new
    resource relied by the current sub-task.
The available agents and the corresponding descriptions are: [code_agent: Generate code in
    Python for precise computations to solve the given task. math_agent: Answer math
    questions by reasoning step-by-step. search_agent: Call Bing Search API for obtaining
    information regarding the given task. commonsense_agent: Answer the given question using
    commonsense reasoning.].
Given the user query: %s, the preliminary task decomposition is: %s.
But the sub-task: %s cannot be solved only with agent %s. Now you are responsible for planning
     this sub-task in detail and choose the most suitable agents based on agents'
    capabilities. Output the new sub-tasks with the format above directly. Make sure that
    there are no duplicate content between new sub-tasks and the given preliminary task
    decomposition.
'''
```

Figure 5: Prompt for planning in detail.

## A.4 RE-DESCRIBE

The prompt for re-describing the sub-task is shown in Figure 6.

```
redescribe_subtask_prompt = '''
Rewrite the following sentence based on the given example, while keeping the key information
    unchanged. Besides, output the rewritten sentence in the form like ***rewritten***.
---
Here is an example:
Example sentence: 'Determine the population of the United States in 2022.'
Sentence to be rewritten: 'Assess the population of China in 2022.'
Rewritten sentence: 'Determine the population of China in 2022.'
Output: ***'Determine the population of China in 2022.'***
---
Example sentence: %s
Sentence to be rewritten: %s
'''
```

Figure 6: Prompt for re-describing the sub-task.

## A.5 DETECTOR

The prompt for the detector is shown in Figure 7.

```
plan_detector_prompt = '''
You are a plan detector responsible for analyzing the completeness and redundancy of the plan.
     Given the query and the plan formulated to solve the query, which involves several sub-
     tasks, you should do the following things:
1. **Detect whether the plan satisfies the completeness.**: Evaluate whether the set of
     subtasks covers all key aspects of the original task including important numbers and
     nouns. Specifically, check if each important element and requirement from the original
     task is addressed by at least one subtask. Provide a brief explanation if any key
     information is missing.
2. **Detect whether the plan satisfies the non-redundancy.**: Evaluate whether any two sub-
     tasks contain identical information and requirements. If there is any redundant part,
     list and provide suggestions for optimizing the plan.
---
For example:
Task: If a plane can carry 300 passengers and flies from Brazil to Nigeria with a full load,
     then returns with only 75% capacity filled, how many passengers in total has it
     transported between the two countries in one round trip?
Subtask 1: Determine the number of passengers transported from Brazil to Nigeria in one flight
     with a full load.    Dependency: []
Subtask 2: Determine the number of passengers transported from Nigeria to Brazil in one flight
     with 75% capacity filled.    Dependency: []
Subtask 3: Calculate the total number of passengers transported between Brazil and Nigeria in
     one round trip.    Dependency: [1, 2]
Analyse: This plan does not satisfy completeness because the subtask loses the information of
     'a plane can carry 300 passengers' of the original task. This plan satisfies non-
     redundancy because each subtask has a unique focus and there is no overlap in the
     information covered.
Suggestions: Add the information of 'a plane can carry 300 passengers' to subtask 1 and
     subtask 2.
---
If there is no need to modify the plan, just return 'The plan satisfies completeness and non-
     redundancy.'.
'''
```

Figure 7: Prompt for the detector.

## A.6 EVALUATION

The prompt for comparing the response with the ground truth is shown in Figure 8.

```
evaluate_prompt = '''
You are CompareGPT, a machine to verify the correctness of predictions. Answer with only yes/
     no.
You are given a question, the corresponding ground-truth answer and a prediction from a model.
     Compare the "Ground-truth answer" and the "Prediction" to determine whether the
     prediction correctly answers the question. The prediction may contain extra information,
     but a correct prediction includes the ground-truth answer. You can answer "yes" if the
     prediction includes the ground-truth answer. You must answer "no" if there are any
     specific details in the ground-truth answer that are not mentioned in the prediction. If
     the prediction states something as a possibility, treat it as a definitive answer. Note
     that the error within three decimal places is negligible.
---
Question: %s
Ground-truth answer: %s
[Start of the prediction]
%s
[End of the prediction]
'''
```

Figure 8: Prompt for comparing the response with the ground truth.

## A.7 AGENTS

The prompts for the code, math, search and commonsense agent are shown in Figure 9, 10, 11, 12 separately. Specifically, the commonsense agent can generate commonsense information like the metal melting points and the atomic numbers of helium and hydrogen.

```
code_agent_prompt = '''You ara a code agent. You can be used for : 1) computing large numbers,
    fractions or decimals. 2) counting or averaging long lists of numbers. 3) performing
    date-related operations, such as counting the number of days between two dates. Write
    code in Python to solve the given task with history. Give the code in the following form
    directly.
- Here is an example:
Task: Calculate the combined population of China and India in 2022.
History: The answer of 'Determine the population of China in 2022' is 1.412B. The answer of '
    Determine the population of India in 2022' is 1.417B.
Code:
```python
# Given populations
population_china_2022 = 1.412 * 10**9  # 1.412 billion
population_india_2022 = 1.417 * 10**9  # 1.417 billion

# Calculate combined population
combined_population_2022 = population_china_2022 + population_india_2022

# Print the result
print(f"The combined population of China and India in 2022 is {combined_population_2022}
    people.")
```
---
Task: %s
History: %s
Code:
'''

rewrite_code_agent_prompt = '''You are a rewrite agent. Given the input question, the code
    addressing this question and the corresponding output, rewrite the output into a complete
    sentence that integrates information from the question and the code output.
---
Question: %s
Code: %s
Code output: %s
Output:
'''
```

Figure 9: Prompt for the code agent.

```
math_agent_prompt = '''You are a math agent. You can answer math questions by reasoning step-
    by-step with the data provided in the question and history. Present the answer "ANS" to
    the subquestion in LaTeX using the format 'The answer is \boxed{ANS}.' without any units
    in the box.
---
Question: %s
History: %s
Solution:
'''
```

Figure 10: Prompt for the math agent.

```
search_agent_prompt = '''You are a search agent. Write a concise, informative Bing Search
    query for obtaining information regarding the given task.
- Here is an example:
Task: Determine the population of China in 2022.
History: None
Search query: China population 2022
---
Task: %s
History: %s
Search query:
'''
rewrite_search_agent_prompt = '''You are a rewrite agent. Given the search question, the
    response in the web_pages from the bing search api, answer the search question with the
    information from the response in concise words. Remove redundant information that is
    irrelevant to the question.
---
Question: %s
Answer_box: %s
Answer:
'''
```

Figure 11: Prompt for the search agent.

```
commonsense_agent_prompt = '''You are a commonsense agent. You can answer the given question
    with logical reasoning, basic math and commonsense knowledge.
---
Question: %s
History: %s
Solution:
'''
```

Figure 12: Prompt for the commonsense agent.

## B FAULT TOLERANCE: UNEXPECTED CHANGES IN AGENT AVAILABILITY

We categorize changes in agent availability into two situations: those occurring during non-execution periods and those during execution.

- *During non-execution periods*: AOP allows for the addition or removal of agents. The meta-agent can first broadcast a simple sync signal to determine agent availability. Only available agents are provided to the meta-agent for task allocation.

- *During execution*: Changes in agent availability during execution indicate that an agent selected for a task may unexpectedly become unavailable, leading to the meta-agent not receiving a response to this sub-task. In such scenarios, the meta-agent is required to reassign the sub-task to another agent with similar capabilities or to further decompose the task (i.e., plan-in-detail).

## C CONSTRUCTING TRAINING DATASET

The prompt for the scorer is shown in Figure 13. The criteria for manual scoring are consistent with the standards outlined in the prompt provided to the scorer.

```
scorer_prompt = '''
Please act as an impartial judge and evaluate the quality of the response provided by the %s
    to the user task. Your evaluation should consider three factors: correctness, relevance
    and completeness. Assign a score of 0, 1 or 2 for each factor and provide a brief
    explanation for your score. The following is the grading criteria.
---
Correctness
0: The response contains severe errors and is completely inaccurate.
1: The response has some errors, but the main content is generally correct
2: The response is completely accurate and fully meets the requirements of the task.
Relevance
0: The response is minimally relevant to the task and completely off-topic.
1: The response is somewhat relevant to the task but may include some unrelated content.
2: The response is highly relevant to the task, directly addressing the core issue without any
    unrelated content or deviation.
Completeness
0: The response lacks necessary detail or key information, resulting in an incomplete
    understanding of the task.
1: While the response addresses part of the task, more information or content is needed for
    completeness.
2: The response provides comprehensive information and detailed explanations.
---
Besides, summarize the final result in the form like '**Correctness: score, Relevance: score,
    Completeness: score**' at the end of your response, where score can be chosen from 0, 1
    and 2.
---
Task
%s
[The Start of Agent's Response]
%s
[The End of Agent's Response]
'''
```

Figure 13: Prompt for the scorer.

We design a mapping function that converts different combinations of correctness, relevance, and completeness values into a final score, which is shown in Figure 14. Then, a score of 5 or higher is considered sufficiently high, while a score of 1 or lower is considered sufficiently low.

```
score_map = {
    (2, 2, 2): 8,
    (2, 1, 2): 7,
    (2, 2, 1): 6,
    (2, 1, 1): 5,
    (1, 2, 2): 4,
    (1, 1, 2): 3,
    (1, 2, 1): 2,
    (1, 1, 1): 1,
}
def level_score(correctness, relevance, completeness):
    return score_map.get((correctness, relevance, completeness), 0)
```

Figure 14: Mapping function.

# D EXPERIMENTS ON MORE DATASETS

## D.1 PERFORMANCE ON DROP AND IIRC

In addition to Husky-QA (Kim et al., 2024), we also conduct experiments on DROP (Dua et al., 2019) and IIRC (Ferguson et al., 2020). The experimental results are shown in Table 4, which indicate that AOP significantly outperforms the baselines, achieving at least a 5.0% improvement on both DROP and IIRC. These experimental results further confirm the effectiveness and advancements of AOP.

Table 4: Accuracy (%) comparisons between AOP and baselines on DROP and IIRC.

| Method | DROP | IIRC |
|---|---|---|
| GPT-4o | 23.0 | 33.0 |
| CoT | 26.0 | 35.0 |
| Zero-Shot CoT | 24.5 | 33.0 |
| Meta-Agent | 25.5 | 32.5 |
| Meta-Agent: Traversal | 27.5 | 34.0 |
| REACT | 29.0 | 36.0 |
| HUSKY | 28.0 | 36.5 |
| AOP | **34.0** | **41.5** |
| AOP (reward model trained on Husky-QA) | 32.0 | 39.0 |

### D.2 GENERALIZATION CAPABILITY OF THE REWARD MODEL

To evaluate the generalization capability of the reward model, we train a reward model based on Husky-QA and utilize it in experiments on the DROP and IIRC datasets. As shown at the bottom in Table 4, the reward model trained on one dataset demonstrates good generalization to other datasets (though it experiences a slight performance drop compared to the specifically trained reward model), achieving superior performance compared to baselines.

## E FURTHER INVESTIGATION OF EXPERT AGENTS

### E.1 EXPERT AGENTS

To investigate the effectiveness of expert agents powered by LLMs fine-tuned on task-specific datasets, we adopted Qwen2-Math-7B (Yang et al., 2024) as the backbone LLM for the math agent and DeepSeek-Coder-V2 (Zhu et al., 2024) as the backbone LLM for the code agent. The experimental results show that integrating these expert agents results in an additional 1.8% improvement on Husky-QA compared with using GPT-4. These results demonstrate the potential for further enhancements of AOP by incorporating more powerful expert agents.

### E.2 MULTIPLE AGENTS WITH SAME EXPERTISE

Based on AOP, we set up an experiment involving 4 different math agents, using GPT-3.5, GPT-4o, Qwen2-Math-7B, and Llama-3.2-3B, respectively. We instruct the multi-agent system to solve the complex math problem from MATH (Hendrycks et al., 2021). With AOP, the queries are decomposed into multiple queries and these agents run in parallel to resolve them. The experimental results are shown in Table 5, demonstrating the effectiveness (at least 6% improvements) of AOP.

Table 5: Accuracy (%) comparisons on MATH.

| | GPT-3.5 | GPT-4o | Qwen2-Math-7B-Instruct | Llama-3.2-3B | AOP |
|---|---|---|---|---|---|
| MATH | 43 | 62 | 66 | 36 | **72** |

## F ADDITIONAL EXPERIMENTAL RESULTS AND DISCUSSIONS

### F.1 QUANTITATIVE EVALUATION ON SOLVABILITY, COMPLETENESS, AND NON-REDUNDANCY

We conduct a quantitative evaluation to demonstrate the effectiveness of AOP in terms of solvability, completeness, and non-redundancy. Specifically, we compare the decomposed sub-tasks provided by AOP to those provided by GPT-4o. These sub-tasks are assessed by an LLM-based evaluator,

providing binary scores for solvability, completeness, and non-redundancy, with scores of 1 for those that meet these principles and 0 for not. The experimental results (averaged scores), as summarized in Table 6, show that AOP achieves significant improvements.

Table 6: Quantitative Evaluation on Solvability, Completeness and Non-redundancy.

|  | Solvability | Completeness | Non-redundancy |
|---|---|---|---|
| GPT-4o | 0.763 | 0.822 | 0.986 |
| AOP | 0.938 | 0.969 | 0.993 |

### F.2 EFFECT OF DIFFERENT REPRESENTATION APPROACHES FOR AGENT DESCRIPTION

In this study, we employ a combination of predefined natural language descriptions and representative works as the representations of agents descriptions. A natural language description can be manually provided or automatically generated, detailing the general and task-independent abilities of agents. While providing a comprehensive and detailed natural language description can be beneficial, it also requires effective prompt engineering. The representative (please refer to Sec. 4.3 for more details) works complement the natural language descriptions and are often task-dependent, allowing for continuous updates during execution.

We conducted experiments on Husky-QA to compare the effectiveness of different representation approaches. The results, shown in Table 7, indicate that using natural language descriptions achieves significantly superior performance compared to using only representative works, which motivates the majority of existing studies to adopt natural language descriptions. Besides, incorporating task-specific representative works on top of natural language descriptions leads to a further 11.9% performance boost, demonstrating the effectiveness of our proposed combined representation approach.

Table 7: Comparisons among Natural Language Descriptions and Representative Works.

| Approaches | Accuracy (%) |
|---|---|
| Natural Language Descriptions | 31.8 |
| Representative Works | 16.8 |
| Both | 43.7 |

Besides, in this study, we employ the natural language descriptions adapted from Husky Kim et al. (2024) to ensure fair comparisons. To further investigate the effect of natural language descriptions, we provide two different versions: (i) LLM-generated: We instruct the LLM to generate the natural language descriptions based on how we describe the agents in Section 5.1; (ii) Expert-written: The natural language descriptions crafted by a human expert. The comparisons on Husky-QA are shown in Table 8. From the table, we can observe that using simple, fully automated natural language descriptions may somewhat affect the overall performance. Besides, incorporating more expert insights into the generation of natural language descriptions can provide an additional performance boost. These experimental results align with the current intuitive understanding of prompt engineering.

Table 8: Comparisons among Different Natural Language Descriptions.

|  | Accuracy (%) |
|---|---|
| Descriptions adapted from Husky | 43.7 |
| LLM-generated descriptions | 40.4 |
| Expert-written descriptions | 44.2 |

### F.3 THE EFFECTIVENESS OF DETECTOR IN HANDLING MISSING DEPENDENCIES

We conduct an experiment to evaluate the effectiveness of the detector in handling missing dependencies. To be more specific, the detector is instructed to determine whether there are any missing dependencies before a sub-task is assigned to an agent for execution. If such dependencies exist, the execution results of these dependent sub-tasks must also be provided as inputs. The adopted prompt is provided in Figure 15. If there exists any missing information, the meta-agent will add sub-tasks to complete the dependencies. The experiments conducted on Husky-QA show that, by enabling the detector to handle missing dependencies, AOP obtains a 2.5% performance increase in accuracy (from 43.7% to 46.2%).

### F.4 THE LIMITATIONS ON THE NUMBER OF AGENTS

AOP has no inherent limitations on the number of agents. However, as the meta-agent is powered by LLMs, increasing the number of agents is subject to the contextual window length restrictions of LLMs (descriptions of a large number of agents could exceed the LLM's output length limit, such as 128K tokens), and the effectiveness of LLMs could be affected by the ability to handle long contexts.

In practical applications, it is important to consider that as the number of agents increases, there may be redundant and functionally similar agents, which motivates us to apply some extension strategies to handle. For example, agents can be grouped according to their abilities. When the meta-agent allocates sub-tasks, it can first select an agent group for each sub-task and then further choose the most suitable agent within the group. This strategy respects the LLM's context window limits and enhances the accuracy of agent selection.

```
dep_detect_prompt = '''
You are an intelligent detector tasked with determining whether the provided dependency
    information is sufficient to complete a given task. If the dependency information is
    insufficient, you will review all historical data to find supplemental information. If
    neither the dependency information nor the historical data is sufficient, you will
    identify and list the missing information. Besides, the information in the task or a
    given query can be viewed as available in the dependency information.

Input:
Task: {Task description}
Dependency Information: {Dependency information provided}
Historical Data: {Historical data (listed with serial numbers)}
Available query: {The query whose information can be viewd as available inthe dependency
    information.}

Requirements:
1. Assess Dependency Information: Evaluate if the provided dependency information is
    sufficient to complete the task. If sufficient, answer ***Yes***; if not, answer ***No
    ***. Besides, if the answer is ***Yes***, there is no need to check the following
    requirements.
2. Review Historical Data: If the answer above is ***No***, check the historical data to
    identify any relevant entries that can supplement the task requirements. List the serial
    numbers of the relevant entries as ˜˜˜numbers˜˜˜.
3. Identify Missing Information: If the historical data also cannot supplement the missing
    details, explicitly list what information is still required to complete the task as
    $$$required additional information$$$.

Output Format (strictly follow this structure):
1. Is the dependency information sufficient: ***Yes***/***No***
2. Relevant information from historical data: ˜˜˜numbers˜˜˜ or ˜˜˜None˜˜˜
3. Missing information: $$$Specific missing information$$$ or $$$None$$$

Examples:
---
Input:
Task: Calculate the total population of China and the United States in 2022.
Dependency Information:
China's population in 2022 is 1.4 billion.
Historical Data:
1. China's population in 2022 is 1.4 billion.
2. The United States population in 2022 is 330 million.
3. Total world population in 2022: 8 billion.
Available query: If the populations of China and India were combined in 2022, how many
    countries with a population of 70,850,000 each could be formed from this total population
    without leaving anyone out?
Output:
1. Is the dependency information sufficient: ***No***
2. Relevant information from historical data: ˜˜˜2˜˜˜
3. Missing information: $$$None$$$
---
Input:
Task: Calculate the total population of China and the United States in 2022.
Dependency Information:
China's population in 2022 is 1.4 billion.
Historical Data:
1. China's population in 2022 is 1.4 billion.
2. Total world population in 2022: 8 billion.
Available query: If the populations of China and India were combined in 2022, how many
    countries with a population of 70,850,000 each could be formed from this total population
    without leaving anyone out?
Output:
1. Is the dependency information sufficient: ***No***
2. Relevant information from historical data: ˜˜˜None˜˜˜
3. Missing information: $$$The United States population in 2022$$$
----
Input:
Task: %s
Dependency Information:
%s
Historical Data:
%s
Available query: %s
Output:
'''
```

Figure 15: Prompt for the detector in handling missing dependencies.

