# OpenReview forum: "Agent-Oriented Planning in Multi-Agent Systems"
_ICLR.cc/2025/Conference — ICLR 2025 Poster_

### Official Review · Reviewer_AMcZ · 2024-10-25

**Soundness:** 3
**Presentation:** 3
**Contribution:** 2
**Rating:** 8
**Confidence:** 3

**Summary:**

This paper presents a new framework for multi-agent systems where meta-agents decompose user queries into sub-tasks, allocate these tasks to appropriate agents, and evaluate the results based on three key design principles: solvability, completeness, and non-redundancy. The meta-agent also uses a reward model to assess agent performance and includes a feedback loop for continuous improvement. The proposed framework demonstrates noticeable advancements in solving complex real-world problems by efficiently coordinating agents compared to existing methods.

**Strengths:**

This paper proposes an interesting framework for agent-oriented planning in multi-agent systems, including an integrated feedback loop and reward model, which appear to be novel.

This paper is well-structured, with a clear methodology and comprehensive experimental analysis.

This paper is clearly written and well-organized, making the complex concepts of agent-oriented planning accessible.

This paper addresses a crucial challenge in multi-agent systems—optimizing task decomposition and agent coordination, which has substantial implications for practical applications in artificial intelligence.

**Weaknesses:**

While this paper explored a very important problem of agent-oriented planning, I do have a few questions/concerns regarding the reported research work:

1. The principles of solvability, completeness and non-redundancy appear to be very general in nature. They may not be entirely new to the field of multi-agent systems or task planning. The novelty may not lie in the principles themselves but in how they are implemented within the specific context of large language models (LLMs) and agent-oriented systems. While the paper integrates these principles into the implementation of a meta-agent, the principles alone may not represent a breakthrough. This paper could benefit from a more detailed discussion of how these principles are uniquely applied in the context of LLM based multi-agent systems.

2. There seems to be a clear lack of technical details regarding how each agent is constructed, operates, and integrates with the meta-agent. While the paper describes agents such as the “search agent,” “math agent,” and “commonsense agent,” I don't quite understand their underlying architectures and decision-making processes (e.g., the underlying models used for each agent and their training processes). Additionally, the descriptions of agent capabilities seem to be vague. This lack of technical depth makes it difficult to assess the robustness, scalability, and adaptability of the agents, as well as their performance in diverse real-world applications. Without these technical details, the framework’s effectiveness remains partially speculative, relying more on theoretical design principles than on concrete, replicable methods.

3. The reward model developed in this paper operates like a surrogate to predict the performance of using any specific task agent to tackle any given sub-task. The general idea of predicting agents' performance is not new. The concrete design of the network architecture and the training algorithm for the reward model may be new. However, the corresponding technical contribution was not clearly highlighted and justified. The same concern also applies to the feedback loop, which is commonly used to update the surrogate models in the literature. Its novelty under the newly proposed framework may need to be better elucidated. Potentially, this paper may benefit by providing a more detailed comparison with existing surrogate models or a clearer explanation of how their approach differs from standard feedback loops in the literature.

4. This paper introduces multiple algorithm/system parameters, such as reward model thresholds, agent selection criteria, and similarity measures, which may be challenging to fine-tune. This complexity makes it potentially difficult to adapt the proposed multi-agent system to new datasets or applications, as optimal tuning may require deep domain-specific knowledge and extensive experimentation, further affecting the system’s usability. In line with this concern, it might be helpful for the authors to provide some concrete guidelines or heuristics for parameter tuning, or to discuss potential strategies for automating or simplifying this process for new datasets or applications.

5. The experimental evaluation in the paper appears to be limited in scope, as it primarily focuses on a single numerical reasoning dataset. It remains questionable whether the newly developed multi-agent system can generalize well across diverse real-world tasks or more complex, domain-specific applications. Hence, the authors may need to discuss potential challenges in applying their system to more diverse or complex tasks.

**Questions:**

Could you elaborate on how the principles of solvability, completeness, and non-redundancy are uniquely adapted for LLM-based multi-agent systems in this framework?

Could you clarify the unique aspects of your reward model and feedback loop in this framework? How do they differ from traditional surrogate models, and what specific contributions do they offer to agent-oriented planning in LLM-based systems?

What strategies or guidelines do you suggest for fine-tuning the system’s parameters for new datasets or applications? Are there ways to simplify or automate this tuning process to improve usability?

**Details Of Ethics Concerns:**

No ethical concerns.

---

> ### Author Response · Authors · 2024-11-25
>
> We sincerely appreciate your detailed comments and valuable suggestions! We provide the following responses to address your concerns and answer your questions point by point.
>
> > W1 \& Q1: This paper could benefit from a more detailed discussion of how these principles are uniquely applied in the context of LLM based multi-agent systems.
>
> **Responses**: Thank you for your detailed comments and helpful suggestions!
> - Designing an agent-oriented planning framework that adheres to these principles is indeed non-trivial and brings unique challenges. The reasons are twofold. Firstly, the meta-agent has a limited understanding of the abilities of expert agents in the multi-agent systems, which can significantly impact how effectively the meta-agent decomposes and allocates tasks. Secondly, LLM-powered agents, including both expert and meta-agents, are still far from perfect.
> - To apply these principles in the context of LLM-based multi-agent systems, the proposed framework includes a fast decomposition and allocation process to generate initial planning. Such initial planning would be continuously refined and improved according to the detector and scorer to satisfy solvability, completeness, and non-redundancy. And a feedback loop is also established to enhance the representations of the abilities of expert agents.
> - While these principles may appear general in nature, our work, to the best of our knowledge, **is the first one that proposes explicit specification and systematic design that follows these principles** in the domain of LLM-based multi-agent systems. We hope that these principles, along with the framework we have outlined, can serve as a good standing point and inspire further research in this field.
>
> The above discussions have been incorporated into Section 3 of the revised paper that has been uploaded. Thank you again for your suggestions for improving our submission.
>
>
>
> > W2: There seems to be a clear lack of technical details regarding how each agent is constructed, operates, and integrates with the meta-agent... Without these technical details, the framework’s effectiveness remains partially speculative, relying more on theoretical design principles than on concrete, replicable methods.
>
> **Responses**: Thank you for your comments. For more technical details regarding agents:
> - *How agents are constructed and operate*: LLM-powered agents are constructed by providing specialized tools and unique system prompts that set their identity and instructions. These agents are powered by LLMs for query understanding, tool use, and response generation. For example, a search agent would be enabled with search engine APIs and instructed to call these APIs to obtain up-to-date information, while the code agent would have access to the code interpreter and be instructed to write and execute code for solving problems.
> - *How agents integrate with the meta-agent*: The meta-agent serves as the brain and controller of the multi-agent system, which allocates subtasks to the agents by sending messages. Once an agent completes its task, it returns the results to the meta-agent in a similar message flow.
> - *The descriptions of agent capabilities*: In this study, we employ a combination of predefined natural language descriptions and representative works. A natural language description can be manually provided or automatically generated, detailing the general and task-independent abilities of agents. The representative works consist of tasks that the agent has effectively tackled in the past, which complement the natural language descriptions and are often task-dependent, allowing for continuous updates during execution.
> - *Replicable*: All technical details, including the adopted system prompts and the LLMs, are included in Appendix A and B in the submission. Besides, we will release the source code to promote further research in the community.
>
> We have incorporated the above technical details and made them more clear in the revised paper. Thank you again!

---

> ### Author Response · Authors · 2024-11-25
>
> > W3.1: The concrete design of the network architecture and the training algorithm for the reward model may be new. However, the corresponding technical contribution was not clearly highlighted and justified.
>
> **Responses**: Thank you for your comments on the reward model.
> - Different from existing studies that use a reward model to evaluate generated responses, the proposed framework utilizes the reward model to predict the quality of agents' responses to tasks for assessing whether an agent is suited for a task, providing an efficient evaluation method that does not need the execution of tasks.
> - To achieve this, the construction of the reward model's training data differs from that of previous studies. Specifically, we gather sub-tasks and agent descriptions as inputs and use the scores of generated responses as outputs to form the training data. The training objectives and detailed training process are elaborated in Section 4.2 of the paper.
> - We have conducted experiments to measure the contributions of the reward model (RM). The results are presented in Table 2 and Table 3 of the paper, and summarized as follows. From the table we can observe that, the proposed framework experiences a 5\% performance drop when the reward model is removed. Besides, an RM trained with manual scoring and comprehensive parameter tuning can further lead to significant improvements (3.2\% and 4.2\%, respectively). These results demonstrate the importance and contributions of the reward model.
> | | Accuracy |
> | ----- | ---- |
> | Ours | 43.7\% |
> | w/o RM | 38.7\% (-5.0\%) |
> | w/ RM trained based manual scoring | 46.9\% (+3.2\%) |
> | w/ full parameter tuning RM | 46.9\% (+4.2\%) |
>
> We have incorporated the above discussions on the reward model and made them more clear in the revised paper. Hope these responses can address your concerns about the reward model. Thank you again!
>
>
>
> > W3.2: Potentially, this paper may benefit by providing a more detailed comparison with existing surrogate models or a clearer explanation of how their approach differs from standard feedback loops in the literature.
>
> **Responses**: Thank you for your suggestions on the feedback loops.
> - To the best of our knowledge, there is currently no existing feedback loop that can be directly applied to multi-agent systems for agent-oriented planning.
> - We conduct an additional ablation study to measure the contribution of the feedback loop. The experimental results indicate that the proposed framework experiences approximately a 1\% performance decrease when the feedback loop is disabled.
>
> We have incorporated the above discussions and experiments in Appendix E.8 of the revised paper. Thank you again.
>
>
> > W4 \& Q3: it might be helpful for the authors to provide some concrete guidelines or heuristics for parameter tuning, or to discuss potential strategies for automating or simplifying this process for new datasets or applications.
>
> **Responses**: Thank you for your comments on the hyperparameter tuning.
> - In this study, the only hyperparameter that needs tuning is the threshold used for the reward model to define a sufficiently good plan (i.e., the subtask and the corresponding assigned agent). This score can be provided by a human-expert scorer or an LLM-based scorer, combining aspects such as correctness, relevance, and completeness. These scores can be flexibly set up according to downstream tasks, and, for convenience, normalization can be applied.
> - The setting of the threshold also depends on the capability of the LLM, particularly its instruction-following ability. A powerful LLM might allow for more error tolerance, meaning that it can still provide satisfactory responses even if the plan might not get a rather high score. When the expert agent's capabilities are not as powerful, the thresholds need to be set higher (for a good enough plan), which may lead to more iterations needed to refine plans.
>
> We have added the above discussions on the hyperparameter tuning in Appendix F.3 of the revised paper. Thank you again for your comments!

---

> ### Author Response · Authors · 2024-11-25
>
> > W5: The experimental evaluation in the paper appears to be limited in scope, as it primarily focuses on a single numerical reasoning dataset.
>
> **Responses**: Thank you very much for your suggestions on adding experiments on more datasets! We conduct additional experiments on two datasets, including DROP[1] and IIRC[2], which are suggested and processed by Husky[3]. The experimental results are shown in the table below, which indicate that the proposed framework significantly outperforms the baselines, achieving at least a 5.0\% improvement on both DROP and IIRC, thereby confirming the effectiveness and advancements of the proposed framework.
> | | DROP (\%)| IIRC (\%)|
> | ----- | ---- | -------- |
> | Gpt-4o | 23.0 | 33.0 |
> | CoT | 26.0 | 35.0 |
> | Zero-Shot CoT | 24.5 | 33.0 |
> | Meta-Agent | 25.5 | 32.5 |
> | Meta-Agent: Traversal | 27.5 | 34.0 |
> | REACT | 29.0 | 36.0 |
> | HUSKY |  28.0 | 36.5 |
> | Ours | **34.0** | **41.5** |
>
> We have added the above experimental results to Appendix E.1 in the revised paper that has been uploaded. Thank you once again for your helpful suggestions for further improving our submission.
>
> References:
> [1] DROP: A Reading Comprehension Benchmark Requiring Discrete Reasoning Over Paragraphs.
> [2] IIRC: A Dataset of Incomplete Information Reading Comprehension Questions.
> [3] Husky: A Unified, Open-Source Language Agent for Multi-Step Reasoning.
>
>
> > Q2: Could you clarify the unique aspects of your reward model and feedback loop in this framework? How do they differ from traditional surrogate models, and what specific contributions do they offer to agent-oriented planning in LLM-based systems?
>
> **Responses**: Thank you very much for your comments. Please refer to the above responses to W3.1 and W3.2 for the advances, technical details, and contributions of the reward model and feedback loop in the framework, respectively.
>
>
> ------
>
> Thank you again for your valuable feedback! We have uploaded a **revised paper that includes all the experiments and discussions in the above responses**, with the major modifications clearly highlighted.
> We believe that this submission has been further improved based on your suggestions. We hope these responses can address all your concerns and convince you to lean more towards acceptance of our paper.

---

> > ### Comment · Reviewer_AMcZ · 2024-11-25
> > **Thank the authors for their rebuttals.**
> >
> > I sincerely thank the authors for providing rebuttals for my review. In line with the rebuttals, I have the following related concerns and suggestions that hopefully could be further addressed.
> >
> > Response to Q1:
> >
> > I am interested in empirical studies that examine how initial planning and its continuous refinement can satisfy the principles of solvability, completeness, and non-redundancy. To what extent can each of these principles be fulfilled? How do these principles relate to specific steps in the fast decomposition and allocation process?
> >
> > Response to W2:
> >
> > How much effort should be invested in designing the natural language description, and to what extent does it affect performance? Could you provide additional experimental results to support this? Similarly, how are the representative works selected, and what impact do they have on performance? Additional empirical results would be helpful to substantiate your argument.
> >
> > Response to W4 and Q3:
> >
> > Thanks for the explanation. However, I am still uncertain about the actual effort required for parameter fine-tuning and its practical implications. More experiments may be necessary to address this concern.
> >
> > Response to W3.1:
> >
> > Several existing studies have explored the possibility of predicting the performance of Large Language Models, such as:
> >
> > Zhang, Q., Lyu, F., Liu, X., & Ma, C. (2024). Collaborative Performance Prediction for Large Language Models. arXiv preprint arXiv:2407.01300.
> > Owen, D. (2024). How predictable is language model benchmark performance? arXiv preprint arXiv:2401.04757.
> > Ye, Q., Fu, H. Y., Ren, X., & Jia, R. (2023). How Predictable Are Large Language Model Capabilities? A Case Study on BIG-bench. arXiv preprint arXiv:2305.14947.
> >
> > Hence, I think some previously studied methods (not just those listed above) may serve as alternatives to the reward model. Could additional ablation studies be conducted to demonstrate the advantages of the new reward model compared to existing ones?

---

> > > ### Comment · Reviewer_LMUm · 2024-11-26
> > > **Thank the authors for their rebuttals**
> > >
> > > Like reviewer AMcZ I would like to thank the authors for the rebuttals which I will consider in the discussion.

---

> > > > ### Author Response · Authors · 2024-11-26
> > > > **Thank you very much for your reply**
> > > >
> > > > To Reviewer LMUm:
> > > >
> > > > Thank you very much for your reply!
> > > > Please let us know if these responses meet your expectations. We are eager to engage in further discussions and continue improving our work.

---

> > > ### Author Response · Authors · 2024-11-26
> > > **Thank you very much for your reply**
> > >
> > > To Reviewer AMcZ:
> > >
> > > Thank you very much for your reply! We are working hard to prepare responses to your following related concerns and suggestions. Due to resource constraints, we will provide these responses ASAP.

---

> > > ### Author Response · Authors · 2024-11-27
> > > **Responses to your following questions (1/2)**
> > >
> > > Thank you very much for your reply! We provide the following responses to your questions.
> > >
> > > > To what extent can each of these principles be fulfilled? How do these principles relate to specific steps in the fast decomposition and allocation process?
> > >
> > > **Responses**: Thank you for your comments! The fast decomposition and allocation process generates the initial plans that might not satisfy these principles, which would be continuously refined and improved according to the detector and scorer. We conduct a quantitative evaluation to compare the initial planning and its continuous refinement regarding solvability, completeness, and non-redundancy. The subtasks contained in the plans are assessed by an LLM-based evaluator, providing binary scores for solvability, completeness, and non-redundancy, with scores of 1 for those that meet these principles and 0 for those that do not. The experimental results (averaged scores), as summarized in the following table, demonstrate the significant improvements achieved by the refined process in the proposed framework.
> > > | | Solvability | Completeness | Non-redundancy |
> > > | ----- | ---- | ---- | ---- |
> > > | initial plan | 0.763 |  0.822 |  0.986 |
> > > | refined plan | 0.938 | 0.969 | 0.993 |
> > >
> > > We have added the above experiments in Appendix E.5 in the revised paper. Thank you again!
> > >
> > >
> > > >  How much effort should be invested in designing the natural language description, and to what extent does it affect performance? ... Similarly, how are the representative works selected, and what impact do they have on performance?
> > >
> > > **Responses**: Thank you for your comments on natural language description and representative works.
> > > - Regarding natural language description:
> > >     - In this study, to ensure fair comparisons, we employ the natural language descriptions adapted from Husky [1]. Please refer to Appendix B for detailed descriptions.
> > >     - To further investigate the effect of natural language descriptions, we provide two different versions: (i) LLM-generated: We instruct the LLM to generate the natural language descriptions based on how we describe the agents in Section 5.1; (ii) Expert-written: The natural language descriptions crafted by a human expert. The comparisons on Husky-QA are shown in the following table. From the table, we can observe that using simple, fully automated natural language descriptions may somewhat affect the overall performance, but still outperforms baselines. Furthermore, incorporating more expert insights into the generation of natural language descriptions can provide an additional performance boost. These experimental results align with the current intuitive understanding of prompt engineering.
> > > | | Accuracy (\%) |
> > > | ----- | ---- |
> > > | GPT-4o | 33.3 |
> > > | REACT | 37.6 |
> > > | HUSKY | 39.6 |
> > > | Ours (Descriptions adapted from Husky) | 43.7 |
> > > | Ours (LLM-generated descriptions) | 40.4 |
> > > | Ours (Expert-written descriptions) | **44.2** |
> > >
> > > - Regarding representative works:
> > >     - For each agent, sub-tasks that receive high scores (according to the same threshold) would be selected as its representative works.
> > >     - To further investigate the effect of representative works, we compare two more strategies for selecting representative works, including {\it more representative works} (i.e., lowering the threshold), and {\it w/o representative works} at all. The experimental results are shown in the following table. From the table, we can see that the strategy we adopted for selecting representative works in our paper is effective, and the representative works mechanism achieves robust performance, consistently outperforming baseline methods.
> > > | | Accuracy (\%) |
> > > | ----- | ---- |
> > > | GPT-4o | 33.3 |
> > > | REACT | 37.6 |
> > > | HUSKY | 39.6 |
> > > | Ours | **43.7** |
> > > | more representative works | 40.7 |
> > > | w/o representative works | 31.8 |
> > >
> > > We have added the above experiments and discussions in Appendix E.9 and E.10 in the revised paper. Thank you again!
> > >
> > > References:
> > > [1] Husky: A Unified, Open-Source Language Agent for Multi-Step Reasoning.

---

> > > ### Author Response · Authors · 2024-11-27
> > > **Responses to your following questions (2/2)**
> > >
> > > > I am still uncertain about the actual effort required for parameter fine-tuning and its practical implications. More experiments may be necessary to address this concern.
> > >
> > > **Responses**: Thank you for your comments.
> > > - Note that the proposed framework **does not require any fine-tuning of effort parameters**. The only hyperparameter that needs adjustment is the threshold used by the reward model to define a sufficiently good plan.
> > > - To further explore the effects of the threshold value, we conduct experiments with varying threshold values on the Husky-QA dataset. The experimental results are shown in the following table. These results indicate that the proposed framework has relatively low sensitivity to the hyperparameter, showing that within a reasonable range, the performance only experiences minor changes and remains superior to the baseline.
> > > | | Accuracy (\%) |
> > > | ----- | ---- |
> > > | GPT-4o | 33.3 |
> > > | REACT | 37.6 |
> > > | HUSKY | 39.6 |
> > > | Ours (threshold = 0.875) | 43.7 |
> > > | Ours (threshold = 0.750) | 43.1 |
> > > | Ours (threshold = 0.625) | 42.1 |
> > >
> > > We have added the above experiments and discussions in Appendix F.3 in the revised paper. Thank you again!
> > >
> > >
> > >
> > > > I think some previously studied methods (not just those listed above) may serve as alternatives to the reward model. Could additional ablation studies be conducted to demonstrate the advantages of the new reward model compared to existing ones?
> > >
> > > **Responses**: Thank you for your suggestions on the related studies.
> > >
> > > After reading the recommended papers, we noticed that the methods discussed in these papers might not be well-suited to our scenario. The reasons include (i) In agent-oriented planning for multi-agent systems, the reward model needs to make performance predictions based on the capabilities of the agent, which can vary significantly between different expert agents. (ii) The performance predictions in agent-oriented planning are made on a per-query basis, whereas most of the mentioned studies focus more on the overall effectiveness on an entire benchmark. (iii) Some mentioned studies rely on model family or model size, which might not be available to agents with closed-source LLMs. The proposed framework does not restrict whether the LLM used by an agent is open-source or closed-source.
> > >
> > > These differences in application scenarios make it challenging to conduct a reasonable comparison between the proposed reward model and the recommended studies. Thank you again!
> > >
> > >
> > > ---
> > >
> > > We have uploaded a **revised paper that includes all the experiments and discussions in the above responses**, with the major modifications clearly highlighted. Thank you again for your reply and helpful suggestions!

---

> > > > ### Comment · Reviewer_AMcZ · 2024-11-27
> > > > **Thank the authors for providing further information and experiment results**
> > > >
> > > > Thank the authors for providing further information and experiment results. I think all my major concerns have been addressed. I will raise my score accordingly.

---

> > > > > ### Author Response · Authors · 2024-11-28
> > > > > **Thank you for your reply**
> > > > >
> > > > > We sincerely appreciate your positive feedback and your decision to raise the score! Thank you again for the time and effort you put into reviewing our paper!

---

### Official Review · Reviewer_8PGi · 2024-10-31

**Soundness:** 2
**Presentation:** 2
**Contribution:** 3
**Rating:** 6
**Confidence:** 3

**Summary:**

This paper introduces a framework for agent-oriented planning in multi-agent systems, where a central "meta-agent" decomposes complex user queries into sub-tasks, assigns these sub-tasks to specific agents, and evaluates their performance. The authors focus on three key principles, namely, solvability, completeness, and non-redundancy, to guide efficient task decomposition and assignment. The framework also includes a feedback loop, enabling the meta-agent to refine its planning over time.

The framework’s emphasis on structured decomposition and task dependencies is well-suited for complex queries that involve multiple agents with diverse capabilities. By ensuring each sub-task aligns with an agent’s expertise, the system promotes both efficiency and reliability.

I have a few questions:
- Task dependencies are identified during decomposition, but dependencies may also emerge during execution. How does the framework adapt when unforeseen dependencies between sub-tasks arise? Could this affect the overall solvability or efficiency of the task set?

- The framework depends on structured task decomposition, yet real-world environments are often unpredictable and dynamic. How does this approach handle unexpected changes in agent availability or evolving task requirements without needing a complete re-plan?

- Non-redundancy is a key focus, yet redundancy can be valuable for fault tolerance. How does the framework balance efficiency with the potential need for redundancy, particularly in scenarios where backup solutions could be essential?

- The scalability of the approach isn’t clear to me. As the number of agents and interdependent tasks increase, how does the meta-agent manage complexity? Are there limitations on the scale or number of agents?

- The meta-agent requires detailed knowledge of each agent's abilities for effective task allocation. But it isn’t immediately clear to me, how are these capabilities/descriptions in D represented? Can you provide more details?

- Related to the comment above, can you discuss what representation is best suited for agent descriptions? Are these representations task dependent?

- Another follow-up on the above comment, can you discuss about the process/requirements if these representations need to be verified or even updated through the process? How can the approach be adapted to handle such cases?

- While the feedback loop allows for ongoing improvements, how does it prevent instability or oscillations in planning strategies? How would you introduce safeguards into the system or meta-agent to avoid fluctuating between different planning approaches?

Overall, I'm happy with the contributions made in this paper and vote for a weak accept. I'd be happy to raise my score given mine and my fellow reviewers' comments are adequately addressed. Thanks!

**Strengths:**

-- See above

**Weaknesses:**

-- See above

**Questions:**

-- See above

---

> ### Author Response · Authors · 2024-11-25
>
> We sincerely appreciate your detailed comments and valuable suggestions! We provide the following responses to address your concerns and answer your questions point by point.
>
>
> > Q1: Task dependencies are identified during decomposition, but dependencies may also emerge during execution. How does the framework adapt when unforeseen dependencies between sub-tasks arise?
>
> **Responses**: Thank you very much for the thoughtful comments on the unforeseen dependencies during execution.
> - We agree that unforeseen dependencies can impact the execution of sub-tasks. In the proposed framework, any missing dependencies, whether they arise during decomposition or execution, are considered part of the completeness of subtasks and are identified by a detector. Specifically, before a sub-task is assigned to an agent for execution, the detector is required to determine whether there are any additional dependencies needed beyond those that were identified during decomposition (i.e., unforeseen dependencies). If such dependencies exist, the execution results of these dependent sub-tasks must also be provided as inputs.
> - We also conduct an experiment to evaluate the effectiveness of the proposed improvements of the detector. The results demonstrate a 2.5\% performance increase in accuracy (from 43.7\% to 46.2\%). More experimental settings (such as used prompts) are provided in Appendix E.6 of the revised paper.
>
> We have added the above experimental results and discussions in Section 4.4 and Appendix E.6 to the revised paper that has been uploaded. Thank you once again for your helpful suggestions for further improving our submission.
>
>
> > Q2: How does this approach handle unexpected changes in agent availability or evolving task requirements without needing a complete re-plan?
>
> **Responses**: Thank you very much for the insightful comments on the unexpected changes in agent availability. We categorize changes in agent availability into two situations: those occurring during non-execution periods and those during execution.
> - *During non-execution periods*: The proposed framework allows for the addition or removal of agents. The meta-agent can first broadcast a simple sync signal to determine agent availability. Only available agents are provided to the meta-agent for task allocation.
> - *During execution*: Changes in agent availability during execution indicate that an agent selected for a task may unexpectedly become unavailable, leading to the meta-agent not receiving a response to this sub-task. In such scenarios, the meta-agent is required to reassign the sub-task to another agent with similar capabilities (which relates to responses to Q3 regarding backup agents) or to further decompose the task (i.e., plan-in-detail).
>
> The above discussions on fault tolerance concerning agent availability have been added to Appendix F.1 in the revised paper. Thank you again for your valuable suggestions!
>
>
> > Q3: Non-redundancy is a key focus, yet redundancy can be valuable for fault tolerance. How does the framework balance efficiency with the potential need for redundancy, particularly in scenarios where backup solutions could be essential?
>
> **Responses**: Thank you for your comments on redundancy.
> - On one hand, we agree that adding redundancy among agents is necessary for fault tolerance, especially considering unexpected changes in agent availability (as you mentioned in responses to Q2).
> - On the other hand, we believe that redundancy between sub-tasks should be optimized, which implies that repeated execution of the same operations across sub-tasks should be minimized as much as possible. For example, if the melting point of a particular metal has already been queried in a sub-task, this knowledge should be directly utilized if needed, rather than being queried again in another sub-task.
>
> We have included additional discussions on non-redundancy in Section 3 of the revised paper to make it clearer and more comprehensive. Thank you again for your suggestion!

---

> ### Author Response · Authors · 2024-11-25
>
> > Q4: As the number of agents and interdependent tasks increase, how does the meta-agent manage complexity? Are there limitations on the scale or number of agents?
>
> **Responses**: Thank you for your comments.
> - The proposed framework has no inherent limitations on the number of agents. However, as the meta-agent is powered by LLMs, increasing the number of agents is subject to the contextual window length restrictions of LLMs (descriptions of a large number of agents could exceed the LLM's output length limit, such as 128K tokens), and the effectiveness of LLMs could be affected by the ability to handle long contexts.
> - In practical applications, it is important to consider that as the number of agents increases, there may be redundant and functionally similar agents, which motivates us to apply some extension strategies to handle. For example, agents can be grouped according to their abilities. When the meta-agent allocates sub-tasks, it can first select an agent group for each sub-task and then further choose the most suitable agent within the group. This strategy respects the LLM's context window limits and enhances the accuracy of agent selection.
> - We also conduct an experiment according to the above strategy, which increases the amount of the math agents and code agents to 4 respectively. The performance on Husky-QA is 46.2\%, which is higher than reported results that only involve one math agent and one code agent, due to the addition and management of expert agents. These experimental results show that the above strategy can effectively handle the increasing number of agents.
>
> The above discussions and experiments are added to Appendix F.2 in the revised paper. Thank you again for your helpful suggestions for further improving our submission.
>
>
> > Q5: how are these capabilities/descriptions in D represented? Can you provide more details?
> Q6: Related to the comment above, can you discuss what representation is best suited for agent descriptions? Are these representations task dependent?
> Q7: Another follow-up on the above comment, can you discuss about the process/requirements if these representations need to be verified or even updated through the process? How can the approach be adapted to handle such cases?
>
> **Responses**: Thank you very much for your insightful comments and helpful suggestions on the descriptions of agents. We provide a comprehensive response to all these questions.
> - *Regarding the descriptions of agents*: In this study, we employ a combination of predefined natural language descriptions and representative works as the representations of agents’ descriptions:
>     - A natural language description can be manually provided or automatically generated, detailing the general and task-independent abilities of agents. For example, a description for a code agent can be ``This agent is proficient at writing and executing Python code to solve the tasks''. While providing a comprehensive and detailed natural language description can be beneficial, it also requires effective prompt engineering. In this study, to ensure fair comparisons, we employ simple natural language descriptions for agents similar to those used in previous studies, which can be found in Appendix B.
>     - The representative works consist of tasks that the agent has effectively tackled in the past. These representative works complement the natural language descriptions and are often task-dependent, allowing for continuous updates during execution.
> - *What representation is best*: We conducted experiments on Husky-QA to compare the effectiveness of different representation approaches. The results, shown in the following table, indicate that using natural language descriptions achieves significantly superior performance compared to using only representative works, which motivates the majority of existing studies to adopt natural language descriptions. Besides, incorporating task-specific representative works on top of natural language descriptions leads to a further 11.9\% performance boost, demonstrating the effectiveness of our proposed combined representation approach.
> | | Accuracy |
> | ----- | ---- |
> | Natural Language Descriptions | 31.8\% |
> | Representative Works | 16.8\% |
> | Both | 43.7\% |
>
> - *When representations need to be verified or even updated*: Recent studies[1,2] have explored the update of the natural language descriptions of agents, which are orthogonal to this study and can be a promising future direction. For the representative works, their design inherently supports verification and updates.
>
> The above discussions and experiments on the descriptions are added to Appendix E.7 in the revised paper. Thank you very much for your helpful suggestions!
>
> References:
> [1] Chameleon: Plug-and-play compositional reasoning with large language models
> [2] Self-rag: Learning to retrieve, generate, and critique through self-reflection

---

> ### Author Response · Authors · 2024-11-25
>
> > Q8: While the feedback loop allows for ongoing improvements, how does it prevent instability or oscillations in planning strategies? How would you introduce safeguards into the system or meta-agent to avoid fluctuating between different planning approaches?
>
> **Responses**: Thank you very much for your comments.
> Firstly, when the meta-agent performs task allocations, it starts by referencing the natural language descriptions of agents, as discussed in the responses to Q5-Q7. These descriptions, which outline the general abilities of agents, can help prevent instability to some extent. Secondly, in the feedback loop, we can incorporate some mechanisms, such as comparing the similarity to existing representative works, to help filter out redundant ones or outliers, further ensuring system stability.
>
>
> ------
>
> Thank you again for your valuable feedback! We have uploaded a **revised paper that includes all the experiments and discussions in the above responses**, with the major modifications clearly highlighted.
> We believe that this submission has been further improved based on your suggestions. We hope these responses can address all your concerns and convince you to lean more towards acceptance of our paper.

---

> ### Author Response · Authors · 2024-11-29
> **Thank you and look forward to further discussion**
>
> Dear Reviewer 8PGi:
>
> Thank you for your detailed comments and helpful suggestions! We are wondering if our responses have resolved your concerns. Please let us know if our response and clarification meet your expectations. We are eager to engage in further discussions and continue improving our work.
>
> Best regards,
>
> Authors

---

> ### Author Response · Authors · 2024-12-02
> **Look forward to receiving your feedback on the author responses**
>
> Dear Reviewer 8PGi,
>
> I hope this email finds you well.
>
> We really appreciate your helpful suggestions regarding the unforeseen dependencies, unexpected changes in agent availability, fault tolerance, the descriptions of agents, the feedback loop, and so on. We definitely believe that this submission has been further improved based on your suggestions! As the due of discussion phase is very close, we kindly ask if you could take a moment to review the responses and provide your feedback at your earliest convenience.
>
> Thank you again for the time and effort you put into reviewing our paper!

---

> ### Author Response · Authors · 2024-12-03
> **Look forward to receiving your feedback**
>
> Dear Reviewer 8PGi,
>
> As the discussion phase draws to a close in less than a day, we kindly ask if our responses have addressed your concerns. We look forward to receiving your feedback.
>
> Thank you again for the time and effort you have invested in reviewing our paper!
>
> Best regards,
>
> Authors

---

### Official Review · Reviewer_vz1o · 2024-10-31

**Soundness:** 2
**Presentation:** 3
**Contribution:** 1
**Rating:** 5
**Confidence:** 3

**Summary:**

The paper presents a multi-agent framework for problem-solving. In particular, the framework uses a meta-agent to decompose a task into subtasks and assign each subtask to downstream expert agents based on their expertise. Then, it uses a reward model to evaluate the performance of the expert agents and use the rewards as signals for re-planning. Additionally, the framework incorporates a feedback system to further improve the robustness and accuracy of problem-solving. Empirical analysis over a numerical reasoning dataset indicates how the proposed framework outperforms the benchmarks, such as direct query to GPT-4o and chain-of-thought. Additionally, a set of ablation studies necessitates each component, such as the plan detector and the reward model, within the framework.

**Strengths:**

The paper is well-organized, presenting a detailed and thorough explanation of the proposed framework. A particularly compelling aspect is the approach of breaking down tasks into smaller components and assigning them to specialized agents according to their areas of expertise. This decomposition allows each agent to handle tasks within its domain, enhancing overall efficiency and precision. Additionally, the integration of a feedback system significantly boosts the framework’s performance by refining the agents’ actions and ensuring adaptive learning.

**Weaknesses:**

There are several limitations of the work that the authors may consider to address:

1. Expert agents: the difference between each expert agent is their input prompts, but they are using the same underlying model (GPT-4o). It could be more interesting to replace each expert agent with the current state-of-the-art model in their domain. I believe there are plenty of works to fine-tune the language models for code generation, solving math problems, etc.

2. I don't see the significance of the commonsense agent. Feels like the meta agent is also doing some common sense reasoning (e.g., understanding and decomposing tasks). If possible, please provide some use cases where the commonsense agent is necessary.

3. The cost (time and number of tokens) is high compared to direct querying GPT-4o, given that over 5x cost for only a 10 percent improvement. This probably can be solved by optimizing the expert agents (see point 1).

**Questions:**

1. In Table 1, in my understanding, the prompt tokens and completion tokens refer to the AVERAGE NUMBER of tokens for EACH task, is it correct? Probably it's better to clarify this in the paper.

2. Is this framework capable of collaborating with multiple agents with the same expertise and improving efficiency? For example, using multiple math agents to solve a complex math problem, while these agents run in parallel.

3. In my understanding, the four agents in the experiment are GPT-4o with different input prompts, are there any additional fine-tuning to improve the expertise of each agent?

4. The paper talks about solvability, completeness, and non-redundancy at the beginning of the paper. Are there any quantitative results showing that the proposed framework addressed these challenges?

---

> ### Author Response · Authors · 2024-11-25
>
> We sincerely appreciate your detailed comments and valuable suggestions! We provide the following responses to address your concerns and answer your questions point by point.
>
>
> > W1: It could be more interesting to replace each expert agent with the current state-of-the-art model in their domain.
>
> **Responses**: Thank you very much for your valuable suggestions regarding the expert agents.
> - We replace the math agent with Qwen2-Math-7B[1] and the code agent with DeepSeek-Coder-V2[2], and conduct experiments on Husky-QA. The experimental results are summarized in the following table, demonstrating that the proposed framework is compatible with and can be further enhanced by expert agents.
> | | Accuracy |
> | ----- | ---- |
> | Agents employing GPT-4o | 43.7\% |
> | Expert agents | 45.5\% |
>
> - It is worth noting that the proposed framework does not bring any constraints to LLMs that can be employed as the backbone of agents.
>
> The above discussions and experimental results are added to Appendix E.3 in the revised paper that has been uploaded. Thank you again for your suggestions on the expert agents!
>
>
> > W2: If possible, please provide some use cases where the commonsense agent is necessary.
>
> **Responses**: Thank you for your comments on the commonsense agent. Following the previous study[3], the commonsense agent specializes in solving subtasks that require commonsense knowledge, which is different from what meta-agent does. For example, a subtask ``Determine the melting point of gold and silver'' can be handled by a commonsense agent since it owns the commonsense knowledge of metal melting points.
>
> We have provided better explanations and examples of agents to make them more clear in the revised paper. Thank you again!
>
>
> > W3: The cost (time and number of tokens) is high compared to direct querying GPT-4o, given that over 5x cost for only a 10 percent improvement. This probably can be solved by optimizing the expert agents (see point 1).
>
> **Responses**: Thank you very much for your insightful comments.
> - We agree that using expert agents can, to some extent, reduce the time and number of tokens cost of the proposed framework, since more complex subtasks can be handled by a single expert agent. In the above responses to W1, we have also demonstrated that the proposed framework is compatible with expert agents.
> - In Table 1 of the submission, we aim to highlight that, under a fair comparison, our framework achieves significant performance improvements compared to others, while **the cost of our framework is at the same level compared to existing multi-agent systems and is notably lower than approaches that lacking suitable design**, such as Meta-Agent: Traversal. Note that these complex queries cannot be resolved by a single expert agent alone, which can only complete the subtasks they specialize in.
> - These additional costs, which include task decomposition, allocation, and modifications carried out by the meta-agent, are affordable and **can be worthwhile as long as they bring significant improvements in accuracy and stability when applying real-world applications**. Such an exploration aligns with recent studies [6] in inference time computation, aimed at efficiently utilizing more tokens to resolve tasks that a single inference cannot fulfill.
>
> We have highlighted the above discussions on cost and utility in Section 5.2 in the revised paper. Thank you again!
>
>
>
> > Q1: In Table 1, in my understanding, the prompt tokens and completion tokens refer to the AVERAGE NUMBER of tokens for EACH task, is it correct? Probably it's better to clarify this in the paper.
>
> **Responses**: Thank you for your comments. The prompt tokens and completion tokens refer to the total costs of the whole test data. We have made it clear in the revised paper.
>
>
> > Q2: Is this framework capable of collaborating with multiple agents with the same expertise and improving efficiency? For example, using multiple math agents to solve a complex math problem, while these agents run in parallel.
>
> **Responses**: Thank you for your helpful suggestion!
> Based on the proposed framework, we set up an experiment involving 4 different math agents, using GPT-3.5, GPT-4o, Qwen2-Math-7B-Instruct, and Llama-3.2-3B[4], respectively. We instruct the multi-agent system to solve the complex math problem from MATH[5]. With the proposed framework, the queries are decomposed into multiple queries and these agents run in parallel to resolve them. The experimental results are shown in the following table, demonstrating the effectiveness (at least 6\% improvements) of the proposed framework when applied in the suggested scenario.
> | | Accuracy |
> | ----- | ---- |
> | GPT-3.5 | 43\% |
> | GPT-4o | 62\% |
> | Qwen2-Math-7B-Instruct | 66\% |
> | Llama-3.2-3B | 36\% |
> | ours | 72\% |
>
> We have added the above discussions and experiments in Appendix E.4 in the revised paper. Thank you again!

---

> ### Author Response · Authors · 2024-11-25
>
> > Q3: In my understanding, the four agents in the experiment are GPT-4o with different input prompts, are there any additional fine-tuning to improve the expertise of each agent?
>
> **Responses**: Thank you for your comments. We replace the math agent with Qwen2-Math-7B[1] and the code agent with DeepSeek-Coder-V2[2], and conduct experiments to demonstrate that the proposed framework is compatible with and can be further enhanced by expert agents. Please refer to the responses to W1 for more details.
>
> > Q4: The paper talks about solvability, completeness, and non-redundancy at the beginning of the paper. Are there any quantitative results showing that the proposed framework addressed these challenges?
>
> **Responses**: Thank you very much for your helpful suggestions! We conduct a quantitative evaluation to demonstrate the effectiveness of the proposed framework in terms of solvability, completeness, and non-redundancy. Specifically, we compare the decomposed subtasks provided by our proposed framework to those provided by GPT-4o. These subtasks are assessed by an LLM-based evaluator, providing binary scores for solvability, completeness, and non-redundancy, with scores of 1 for those that meet these principles and 0 for not. The experimental results (averaged scores), as summarized in the following table, show that the proposed framework achieves significant improvements.
> | | Solvability | Completeness | Non-redundancy |
> | ----- | ---- | ---- | ---- |
> | GPT-4o | 0.763 |  0.822 |  0.986 |
> | ours | 0.938 | 0.969 | 0.993 |
>
> We have added these comparisons in Appendix E.5 in the revised paper. Thank you again for further improving our submission.
>
>
> References:
> [1] https://huggingface.co/Qwen/Qwen2-Math-7B-Instruct.
> [2] DeepSeek-Coder-V2: Breaking the Barrier of Closed-Source Models in Code Intelligence.
> [3] Husky: A Unified, Open-Source Language Agent for Multi-Step Reasoning.
> [4] https://huggingface.co/meta-llama/Llama-3.2-3B-Instruct.
> [5] Measuring Mathematical Problem Solving With the MATH Dataset.
> [6] https://openai.com/index/learning-to-reason-with-llms
>
>
> ------
>
> Thank you again for your valuable feedback! We have uploaded a **revised paper that includes all the experiments and discussions in the above responses**, with the major modifications clearly highlighted.
> We believe that this submission has been further improved based on your suggestions. We hope these responses can address all your concerns and convince you to lean more towards acceptance of our paper.

---

> ### Comment · Reviewer_vz1o · 2024-11-26
> **Thanks for the Rebuttal**
>
> Thank you for providing the new results. These results resolve most of my concerns and I would consider increasing the score.
>
> I have a few more questions:
>
> 1. Regarding the feedback loop (Sec. 4.5), my understanding is that the pipeline uses a reward model to obtain a numerical value indicating the quality of the plan. Could you please clarify how to refine the plan (e.g., changing the prompt?) based on a single numerical reward value? Please also let me know if my understanding is incorrect.
>
> 2. Regarding the reward model, if there are some outliers (bad plans) with high rewards, is there a post-processing technique to detect and resolve such outliers?

---

> > ### Author Response · Authors · 2024-11-27
> > **Thank you very much for your reply!**
> >
> > Thank you very much for your appreciation of the above responses! For the additional questions, we provide the following responses:
> > > Could you please clarify how to refine the plan (e.g., changing the prompt?) based on a single numerical reward value?
> >
> > Thank you for your comments. Overall, we need to automatically specify (the most possible) issues of the plan, with the help of the feedback from the reward model and the proposed representative work mechanism, and then instruct the meta-agent to take the appropriate actions to refine the plan.
> >
> > To be more specific, the reward model provides scores to predict the quality of responses when a decomposed subtask is allocated to agents. Based on these scores, the meta-agent would take the following strategies to refine the plan if necessary:
> > - One scenario is when the subtask (generated in the fast decomposition and allocation process) is deemed a failure (i.e., all scores related to this subtask fall below a threshold). In this case, we would instruct the meta-agent to revise and {\it replan} for this subtask (please refer to Appendix A.2 for the adopted prompts).
> > - Another scenario is when allocating the subtask to a particular agent is predicted to be suitable. In this case, we need to determine if the subtask still requires further enhancement, with the help of the representative works of this agent. For example, we might need to supplement details lost during the decomposition ({\it re-describe}, please refer Appendix A.4 for the adopted prompts) or to further decompose the sub-task into simple ones ({\it plan-in-detail}, please refer Appendix A.3 for the adopted prompts).
> >
> > For more detailed and formatted introductions of the above process, please refer to Sections 4.1 and 4.2 in the submission. Thank you again!
> >
> > > Regarding the reward model, if there are some outliers (bad plans) with high rewards, is there a post-processing technique to detect and resolve such outliers?
> >
> > Thank you for your comments on the outliers.
> >
> > In the proposed framework, when an outlier is assigned high rewards, it can still be identified and then refined by the meta-agent through the proposed representative work mechanism. Moreover, the representative works of agents would be continuously enhanced through a feedback loop, improving the description of the agent's task-specific capabilities to address the potential outliers that the reward model may not have handled well.
> >
> > ---
> >
> > Thank you again for your reply and helpful suggestions! Please let us know if these responses meet your expectations. We are eager to engage in further discussions and continue improving our work.

---

> > ### Author Response · Authors · 2024-11-29
> > **Thank you for your reply**
> >
> > We sincerely appreciate your reply and your decision to raise the score! Your positive feedback is invaluable to us, and we are pleased that our responses have addressed your concerns.
> >
> > Thank you again for your time and effort in reviewing our paper!

---

### Official Review · Reviewer_LMUm · 2024-11-03

**Soundness:** 2
**Presentation:** 3
**Contribution:** 2
**Rating:** 3
**Confidence:** 4

**Summary:**

The proposed system uses a LLM to generate a decomposition of a problem into an array of sub tasks that are then allocated to different agents. Then a reward model is used to improve the plan as it is being executed.

**Strengths:**

* The problem of multi-agent planning is highly important with high potential impact.
* How to decompose problems into sub-tasks based on the capabilities of the individual agents is often the key problem.
* Learning the capabilities of other agents is a hard and important challenge.
* The paper is easy to follow.

**Weaknesses:**

* The paper does not relate to the rich and long history of multi-agent planning. A starting point could be [1].
* The design principles seem a bit backwards. Normally, the set of problems that can be solved by a set of agents is defined as the union of the problems that each agent can solve, then you have to check whether the particular problem is within that set. Normally, it is a hard computational problem to determine what problems an agent can solve. If the set of sub-tasks contain redundancy it is either a problem with the planner or there is reason, such as a need to deal with non-deterministic outcomes of actions. To have it as a design principle seems strange.
* In the completeness principle it is mentioned "The array of sub-tasks [...] should include all necessary information" what does this mean? The first principled is defined in terms "resolvable" tasks. The second principle is defined in terms of "necessary information". What is the relation between "necessary information" and "resolvable"?
* It is well known that LLMs cannot generate plans in the sense used in the planning community as there are no guarantees that the plan is correct nor that it actually solves the problem. My impression is thus that the proposed solution is more about learning a model of different agents capabilities than on planning.


[1] Cooperative Multi-Agent Planning: A Survey. Alejandro Torreño, Eva Onaindia, Antonín Komenda, Michal Štolba. ACM Computing Surveys (CSUR), Volume 50, Issue 6.

**Questions:**

* Why is it beneficial to use natural language over PDDL and other languages specifically designed for planning? There are papers on this already [2] so it would make sense to compare to such an approach.
* Similarly, why not represent the agent capabilities in terms of planning operators or some other well-defined formalism rather than (what I presume) is natural language descriptions?
* How do you formally verify that the generated plans are correct and solve the problem?

[2] Translating Natural Language to Planning Goals with Large-Language Models by Yaqi Xie et al.

---

> ### Author Response · Authors · 2024-11-25
>
> We sincerely appreciate your detailed comments and valuable suggestions! We provide the following responses to address your concerns and answer your questions point by point.
>
>
> > W1: The paper does not relate to the rich and long history of multi-agent planning. A starting point could be [1].
>
> **Responses**: Thank you for your valuable suggestions regarding the related work.
> We provide additional discussions related to the studies in Cooperative multi-agent planning as you mentioned:
>
> Cooperative multi-agent planning (MAP) has been an active research area for many years [1]. While early MAP works focus on coordination methods for agents using planning representations [2], a significant turning point is the introduction of MA-STRIPS [3], a minimalist multi-agent extension of the well-known STRIPS planning model [4], which provided a widely accepted standardized format for MAP. Following this, MA-PDDL [5], the multi-agent version of PDDL [6], marked the first attempt to create a de facto standard specification language for MAP tasks. Both symbolic methods and reinforcement learning-based methods [7,8] have become mainstream approaches in MAP. In recent years, the development of LLMs has brought considerable attention to LLM-empowered agents [9]. Some methods have enhanced the planning proficiency of LLMs by incorporating a planner [10-13], while others have explored combining an LLM with a lightweight neural planner [14,15]. These developments have injected new vitality into the advancement of MAP.
>
> The above discussions have been added to the Related Work section in the revised paper that has been uploaded. We appreciate your helpful suggestions for further improving our submission. Thank you again!
>
>
> > W2: The design principles seem a bit backwards. ... it is a hard computational problem to determine what problems an agent can solve. If the set of sub-tasks contain redundancy it is either a problem with the planner or there is reason, such as a need to deal with non-deterministic outcomes of actions.
>
> **Responses**: Thank you for your insightful comments regarding the design principles. Overall, we do not expect the meta-plan to satisfy these principles through a single and simple LLM inference (as you mentioned above, this can be really challenging). Instead, **these principles serve as targets guiding us on how to design strategies and mechanisms to enable LLMs to generate more productive thinking, reasoning, and reflection**, thereby resulting in responses that meet these principles.
> For more detailed responses, please see below:
>
> - As you mentioned above, the problems an agent can solve are hard to determine, especially when relying on a simple inference of LLMs. In the proposed framework, this manifests as uncertainty in the agent's capabilities, leading to the initial plan generated by the fast decomposition and allocation process potentially being unsatisfactory. To address this, we incorporate a reward model to provide feedback signals for the meta-agent to refine the plans and design a representative works mechanism to further enhance the descriptions of agent capabilities.
> - The primary purpose of the non-redundancy principle is to optimize efficiency while ensuring the plans lead to correct responses. Redundancy could be considered a problem with the planner, as it is challenging to ensure the generated plans are non-redundant during fast decomposition. That is why we need to adopt a detector to further refine subtasks. The experiments shown in Table 2 of the submission confirm that the detector can bring a 7.1% improvement.
> - Furthermore, we acknowledge the need to address non-deterministic outcomes of actions in some cases. For instance, initial uncertainty about an agent's capabilities might necessitate redundant calls to ensure subtask execution quality. However, as more tasks are executed and the system's understanding of the agent's capabilities becomes clearer, the system can perform much better in generating plans without redundancy.
>
> We hope these responses address your concerns about the design principles. Thank you again for your comments!

---

> ### Author Response · Authors · 2024-11-25
>
> > W3: In the completeness principle it is mentioned ``The array of sub-tasks [...] should include all necessary information'', what does this mean? What is the relation between ``necessary information'' and ``resolvable''?
>
> **Responses**: Thank you for your comments.
> - *Regarding the completeness principle*: The completeness principle requires that all necessary (i.e., critical) information from the user query be preserved in the decomposed sub-tasks, including essential nouns, quantifiers, and other critical elements. These pieces of information may be distributed across different subtasks. While a subtask might include only some pieces of information from the user query, it is not allowable for any particular piece of critical information to be omitted from all subtasks. The instructions given to the detector for improving the completeness can be found in Appendix A.5 of the paper.
> - *Regarding the relation between necessary information and resolvability*: For a subtask, missing necessary information would make it irresolvable. For example, consider the subtask: "Determine the number of full flights needed to transport 1% of the population of New York." This subtask is not resolvable because it lacks the necessary information given in the user query: “300-passenger capacity”.
>
> We have made the description of the completeness principle more clear in the revised paper. Thank you again for your comments.
>
>
> > W4: It is well known that LLMs cannot generate plans in the sense used in the planning community as there are no guarantees that the plan is correct nor that it actually solves the problem. My impression is thus that the proposed solution is more about learning a model of different agents capabilities than on planning.
>
> **Responses**: Thank you for your comments.
> We agree that LLMs might not always generate reasonable plans with a simple inference, especially without well-designed instructions to promote productive thinking, reasoning, and reflection. This challenge motivates us to propose a framework that uses more inference computation and reward signals to continually refine the plans automatically. While learning different agent capabilities is a critical problem in agent-oriented planning, as discussed in the preliminaries section, enhancing a multi-agent system to effectively perform and refine task decomposition and allocation is also a central focus of the proposed framework, aiming at achieving performance boost compared to systems that rely on simple inference with LLMs.
>
>
> Thank you again for your comments. Hope the above responses about the scope of this paper can well address your concerns.
>
>
> > Q1: Why is it beneficial to use natural language over PDDL and other languages specifically designed for planning?
>
> **Responses**: Thank you for your comments!
> While recent studies propose to leverage external planners and utilize LLMs to transform natural language to other languages like PDDL, several challenges remain, such as:
>
> - PDDL has limited capability in handling continuous actions/states, uncertain effects, or incomplete information. When combining LLMs with PDDL, these methods show mixed performance on tasks with partially specified goals. PDDL and similar languages are typically focused on domain-specific problems like robot path planning. They lack adaptability when addressing more ambiguous, broad questions such as summarizing a company's recent performance. In such cases, we find that natural language question-answering scenarios cannot be effectively formulated within this framework to achieve satisfactory results.
> - LLM-generated PDDL information, particularly goals, is sensitive to natural language prompts. Furthermore, LLMs require sufficiently good examples to generalize well. Poor examples significantly impact LLM-generated results, leading to substantial upfront preparation requirements and limited generalization capabilities, especially when designing task-specific prompts. This phenomenon has been mentioned in works combining LLMs with PDDL [10, 16].
>
> The development of LLMs is progressing rapidly but is still far from perfect. In this study, **we focus more on expressing plans in natural language to better leverage LLMs' understanding and generation capabilities in the natural language**.
>
> Thank you again for your comments. We agree that the combination of LLM and PDDL can achieve excellent results given adequate design, which can be a promising future direction.

---

> ### Author Response · Authors · 2024-11-25
>
> > Q2: Similarly, why not represent the agent capabilities in terms of planning operators or some other well-defined formalism rather than (what I presume) is natural language descriptions?
>
> **Responses**: Thank you for your comments on the descriptions of agent capabilities.
> Regarding LLM-powered agents in this study, different from domains such as robot planning where PDDL excels, it can be challenging to develop a system that fully explains all agents' capabilities using well-defined formalisms. Therefore, to ensure a fair comparison, we adopt similar description approaches as those used in existing studies on LLM-powered agent systems.
>
> We believe that using planning operators or some other well-defined formalism to represent the agent's capabilities is a promising future direction, but it might be out of the scope of this paper. Thank you again!
>
>
>
> > Q3: How do you formally verify that the generated plans are correct and solve the problem?
>
> **Responses**: Thank you for your comments. Regarding the verification of whether the generated plans are correct and solve the problem, we include two types of evaluations:
> - Firstly, we execute the generated plans based on the multi-agent system, which includes problem decomposition and task allocation to the corresponding agents, with refinement during execution. The system then produces a response to the user query. We compare the response to the user query, provided by the system, with the ground truth in the dataset to determine whether the problem has been effectively solved or not. Please refer to Table 1 in the paper for such an end-to-end evaluation.
> - Secondly, we can also directly assess the generated subtasks in terms of solvability, completeness, and non-redundancy. Specifically, we compare the decomposed subtasks provided by our proposed framework to those provided by GPT-4o. These subtasks are assessed by an LLM-based evaluator, providing binary scores for solvability, completeness, and non-redundancy, with scores of 1 for those that meet these principles and 0 for not. The experimental results (averaged scores), as summarized in the following table, show that the proposed framework achieves significant improvements.
> | | Solvability | Completeness | Non-redundancy |
> | ----- | ---- | ---- | ---- |
> | GPT-4o | 0.763 |  0.822 |  0.986 |
> | Ours | 0.938 | 0.969 | 0.993 |
>
> We have added the above results in Appendix E.5 of the revised paper. Thank you again.
>
>
> References:
> [1] Cooperative Multi-Agent Planning: A Survey
> [2] A survey of research in distributed continual planning
> [3] From one to many: Planning for loosely coupled multi-agent systems
> [4] STRIPS: A new approach to the application of theorem proving to problem solving
> [5] A multi-agent extension of PDDL3.1
> [6] PDDL—The planning domain definition language
> [7] Deep reinforcement learning with a natural language action space
> [8] Keep calm and explore: Language models for action generation in text-based games
> [9] A survey on large language model based autonomous agents
> [10] Llm+p: Empowering large language models with optimal planning proficiency
> [11] Dynamic planning with a llm
> [12] Leveraging pre-trained large language models to construct and utilize world models for model-based task planning
> [13] Coupling large language models with logic programming for robust and general reasoning from text
> [14] Keep calm and explore: Language models for action generation in text-based games
> [15] Swiftsage: A generative agent with fast and slow thinking for complex interactive tasks
> [16] Translating Natural Language to Planning Goals with Large-Language Models
>
>
> ------
>
> Thank you again for your valuable feedback! We have uploaded a **revised paper that includes all the experiments and discussions in the above responses**, with the major modifications clearly highlighted.
> We believe that this submission has been further improved based on your suggestions. We hope these responses can address all your concerns and convince you to lean more towards acceptance of our paper.

---

> ### Author Response · Authors · 2024-11-29
> **Thank you and look forward to further discussion**
>
> Dear Reviewer LMUm:
>
> Thank you for your detailed comments and helpful suggestions! We are wondering if our responses have resolved your concerns. Please let us know if our response and clarification meet your expectations. We are eager to engage in further discussions and continue improving our work.
>
> Best regards,
>
> Authors

---

> ### Author Response · Authors · 2024-12-02
> **Look forward to receiving your feedback on the author responses**
>
> Dear Reviewer LMUm,
>
> I hope this email finds you well.
>
> We really appreciate your helpful suggestions regarding the related works in multi-agent planning, the design principles, natural language descriptions, verifications of the proposed framework, and so on. We definitely believe that this submission has been further improved based on your suggestions! As the due of discussion phase is very close, we kindly ask if you could take a moment to review the responses and provide your feedback at your earliest convenience.
>
> Thank you again for the time and effort you put into reviewing our paper!

---

> ### Author Response · Authors · 2024-12-03
> **Look forward to receiving your feedback**
>
> Dear Reviewer LMUm,
>
> As the discussion phase draws to a close in less than a day, we kindly ask if our responses have addressed your concerns. We look forward to receiving your feedback.
>
> Thank you again for the time and effort you have invested in reviewing our paper!
>
> Best regards,
>
> Authors

---

### Official Review · Reviewer_4UEG · 2024-11-04

**Soundness:** 2
**Presentation:** 2
**Contribution:** 3
**Rating:** 6
**Confidence:** 2

**Summary:**

The paper describes a method for solving queries using a multi-agent system, with multiple agents which are experts in at different sub-tasks. A meta-agent is then required to decompose the queries into multiple sub-tasks that are then allocated to suitable agents for solving them. The authors identified three critical principles for task completion:

That is Solvability - Each sub-task should be solvable by at least one of the available agents.

Completeness - The set of sub-tasks should include all necessary information from the original query. That is the aggregation of the sub-tasks should have a comprehensive answer to the original query.

Non-redundancy – The set of sub-tasks should not include duplicate or unnecessary elements.

For each of these requirements, should it fail any, the meta-agent is required to revisit the task decomposition and/or task allocation.

To determine if a sub-task is completely resolved you would need to check if any of the agents can solve a sub-task, but due to the overhead required, they create a training dataset for a reward model using the fast decomposition process and a scorer, which evaluates agent response. This reward model is used to reduce the overhead of agent calls. It is used to approximate whether a sub-task is resolved by an agent by check if its score is sufficiently high (above a certain threshold). In cases where this score is not high enough for any agent, the sub-task is deemed to not fit the solvability criteria and the meta-agent replans.

They then explore when sub-tasks are lacking information/ambiguous or are too complex, and use a similarity calculation over the embeddings of the subtasks. If it is too high, there are tasks similar to it and it should be re-described according to the similar representative  sub-task. If it does not meet a threshold then the sub-task is considered too complex and is modified.
A detector is added to check that all key elements of the original query exist in all the sub-tasks, and that no two sub-tasks resolve the same key elements. If either is failed a recommendation based on that key element is made to help remove overlapping subtasks or supplement missing details.
This is evaluated vs a number of baselines and experiments are done based on a numerical reasoning dataset. It is shown to outperform the baselines.

Ablations are done to show that each component adds meaningfully to the model.

The related work is only listed after all the results.

**Strengths:**

The work shows three components which improve the ability of multi-agent LLM systems to improve the results of general LLM queries as compared to current methods.

All three components are shown to meaningfully add to the overall model, and directly tackle the presented critical principles for design.

**Weaknesses:**

The work is difficult to follow at times, I believe it can be made much clearer, especially with regards to Figure 2. It is difficult to understand this diagram without enough direct context. There is no clear direction that it takes. The detector does not seem to lead to a replan as described.

The ordering of the sections, particularly the related work, and how it reads in context at the end, does not make sense, was this just compiled in the wrong place accidentally?

The results only have one dataset to my understanding. Can this be extended to more datasets?

I have concerns that the rewards model does not generalize outside of this dataset? Or would that simply require more training on bigger datasets?

I believe this work is good and novel, however it could be improved upon with clearer writing.

**Questions:**

The results only have one dataset to my understanding. Can this be extended to more datasets?

I have concerns that the rewards model does not generalize outside of this dataset? Or would that simply require more training on bigger datasets?

The ordering of the sections, particularly the related work, and how it reads in context at the end, does not make sense, was this just compiled in the wrong place accidentally?

Are there cases where the scorers are human experts? It was a bit ambiguous as to whether this was in fact the case.

**Details Of Ethics Concerns:**

Due to ambiguity of human experts with the scoring, I am unable to comment, but believe if that is checked and declared that there are no concerns beyond that.

---

> ### Author Response · Authors · 2024-11-25
>
> We sincerely appreciate your detailed comments and valuable suggestions! We provide the following responses to address your concerns and answer your questions point by point.
>
>
> > W1: I believe it can be made much clearer, especially with regards to Figure 2. It is difficult to understand this diagram without enough direct context.
>
> **Responses**: Thank you for your valuable suggestions regarding Figure 2. We have updated the figure to make it clearer and easier to follow, and we have also added a detailed caption:
>
> Overall architecture of the proposed agent-oriented planning framework. The framework begins with the meta-agent performing a fast decomposition and allocation of the received user query, resulting in an initial plan. A detector is employed to improve the completeness and eliminate redundancy of the plan, while a reward model provides scores to guide the meta-agent in refining the plan further, which involves operations such as replan, plan-in-detail, etc. The refined plan is sent back to the multi-agent system for generating responses to the user query.
>
> These modifications have been incorporated into the revised paper that has been uploaded. We appreciate your helpful suggestions for further improving our submission. Thank you once again!
>
>
> > W2 \& Q3: The ordering of the sections, particularly the related work, and how it reads in context at the end, does not make sense.
>
> **Responses**: Thank you for your suggestions regarding the ordering of the sections. In the revised paper, we have moved the Related Work section to follow the Introduction and precede the Preliminaries section. We believe this arrangement would make the submission more reader-friendly and clear.
>
>
> > W3 \& Q1: The results only have one dataset to my understanding. Can this be extended to more datasets?
>
> **Responses**: Thank you very much for your valuable suggestions! We conduct additional experiments on two datasets, including DROP[1] and IIRC[2], which are suggested and processed by Husky[3]. The experimental results are shown in the table below, which indicate that the proposed framework significantly outperforms the baselines, achieving at least a 5.0\% improvement on both DROP and IIRC, thereby confirming the effectiveness and advancements of the proposed framework.
> | | DROP (\%)| IIRC (\%)|
> | ----- | ---- | -------- |
> | Gpt-4o | 23.0 | 33.0 |
> | CoT | 26.0 | 35.0 |
> | Zero-Shot CoT | 24.5 | 33.0 |
> | Meta-Agent | 25.5 | 32.5 |
> | Meta-Agent: Traversal | 27.5 | 34.0 |
> | REACT | 29.0 | 36.0 |
> | HUSKY |  28.0 | 36.5 |
> | Ours | **34.0** | **41.5** |
>
> We have added the above experimental results in Appendix E.1 in the revised paper that has been uploaded. Thank you once again for your helpful suggestions for further improving our submission.
>
> References:
> [1] DROP: A Reading Comprehension Benchmark Requiring Discrete Reasoning Over Paragraphs.
> [2] IIRC: A Dataset of Incomplete Information Reading Comprehension Questions.
> [3] Husky: A Unified, Open-Source Language Agent for Multi-Step Reasoning.
>
>
>
> > W4 \& Q2: I have concerns that the rewards model does not generalize outside of this dataset? Or would that simply require more training on bigger datasets?
>
> **Responses**: Thanks a lot for your comments. To evaluate the generalization capability of the reward model, we train a reward model based on Husky-QA and utilize it in experiments on the DROP and IIRC datasets. As shown in the table below, the reward model trained on one dataset demonstrates good generalization to other datasets (though it experiences a slight performance drop compared to the specifically trained reward model), achieving superior performance compared to the two strongest baselines.
> | | DROP (\%)| IIRC (\%)|
> | ----- | ---- | -------- |
> | REACT | 29.0 | 36.0 |
> | HUSKY |  28.0 | 36.5 |
> | Ours (dataset-specified reward model) | 34.0 | 41.5 |
> | Ours (reward model trained on Husky-QA) | 32.0 | 39.0 |
>
> We have added the above experimental results in Appendix E.2 to the revised paper. Hope these responses can well address your concerns about the generalization of the reward model.

---

> ### Author Response · Authors · 2024-11-25
>
> > W5: I believe this work is good and novel, however it could be improved upon with clearer writing.
>
> **Responses**: Thank you for your suggestions! We have carefully polished our paper, including re-arranging the content, correcting typos, and enhancing the descriptions to make them more logical and reader-friendly. All these improvements have been incorporated into the revised paper that has been uploaded. Thank you once again!
>
>
> > Q4: Are there cases where the scorers are human experts? It was a bit ambiguous as to whether this was in fact the case.
>
> **Responses**: Thank you for your comments. In this study, we investigate both model-based scorers and human-expert-based scorers, and provide a comparison in Table 3 of the paper. We find that using human-expert-based scorers can lead to further improvements in the overall framework, which can be attributed to their annotations being better aligned with human understanding. However, we predominantly adopt a model-based scorer (using GPT-4o) in most of the experiments, as it is a more cost-effective and generalizable manner, showing that the proposed framework does not heavily rely on human experts.
>
> Thank you again for your comments. We have included the above discussions in Section 5.3 in the revised paper accordingly.
>
>
> ------
>
> Thank you again for your valuable feedback! We have uploaded a **revised paper that includes all the experiments and discussions in the above responses**, with the major modifications clearly highlighted.
> We believe that this submission has been further improved based on your suggestions. We hope these responses can address all your concerns and convince you to lean more towards acceptance of our paper.

---

> > ### Comment · Reviewer_4UEG · 2024-12-02
> >
> > Thank you, the edits you have made do indeed make the paper far more reader friendly.
> >
> > I believe that the further experiments with the DROP and IIRC datasets do add to the idea that this model can generalize outside of the dataset it is specifically trained for, showing it does improve on other methods. This does ease my concerns with regards to the generalization.
> >
> > Thank you for clarifying with regards to the scorers.

---

> > > ### Author Response · Authors · 2024-12-02
> > > **Thank you very much for your reply**
> > >
> > > Thank you very much for your reply and your decision to raise the score! Your positive feedback means a great deal to us, and we are glad that our responses have addressed your concerns.
> > >
> > > We appreciate your time and effort in reviewing our paper. Thank you again!

---

> ### Author Response · Authors · 2024-11-29
> **Thank you and look forward to further discussion**
>
> Dear Reviewer 4UEG:
>
>
> Thank you for your detailed comments and helpful suggestions! We are wondering if our responses have resolved your concerns. Please let us know if our response and clarification meet your expectations. We are eager to engage in further discussions and continue improving our work.
>
>
> Best regards,
>
> Authors

---

> ### Author Response · Authors · 2024-12-02
> **Look forward to receiving your feedback on the author responses**
>
> Dear Reviewer 4UEG,
>
> I hope this email finds you well.
>
> We really appreciate your helpful suggestions regarding Figure 2, the ordering of the sections, experiments on more datasets, the generalization ability of the rewards model, and so on. We definitely believe that this submission has been further improved based on your suggestions! As the due of discussion phase is very close, we kindly ask if you could take a moment to review the responses and provide your feedback at your earliest convenience.
>
> Thank you again for the time and effort you put into reviewing our paper!

---

### Meta-Review · Area_Chair_M8h9 · 2024-12-19

**Metareview:**

The paper concerns solving problems in a multi-agent context, with multiple agents that have different expertise. The authors propose to use an LLM meta agent that decomposes the queries into sub-tasks and allocates the subtasks to the agents.

The reviewers agree that the paper has merit, in particular, appreciating the three meaningful and structured components for task completion.

Most of the weaknesses evolve around the writing, the related work on multi-agent planning, and how the method generalizes. Most of these concerns were mitigated during the discussion phase, and we recommend acceptance of this paper.

**Additional Comments On Reviewer Discussion:**

The discussion was very lively and the authors and reviewers enaged well, improving the paper significantly.

---

### Decision · Program_Chairs · 2025-01-22

Accept (Poster)